# How can embedding models bind concepts?

**Arnas Uselis** [* 1]  **Darina Koishigarina** [* 1]
**Seong Joon Oh** [1 2]

## Abstract

Humans easily determine which color belongs to which shape in multi-object scenes, an ability known as concept binding. Vision–language embedding models such as CLIP struggle with binding: they recognize individual concepts but fail to represent which concepts form which objects. Although CLIP behaves like a bag-of-concepts model in cross-modal retrieval, object information is recoverable from its image and text embeddings separately. We study this tension through the binding function, which maps concepts to scene embeddings. We find that scene embeddings decompose additively into object representations, explaining why uni-modal probes can recover object information. However, CLIP's binding function is high-complexity, which likely prevents the image and text encoders from learning a shared binding mechanism that generalizes to unseen concept combinations. We then ask whether this limitation is fundamental. We show that it is not. In controlled transformer models trained from scratch, binding generalization emerges with sufficient data coverage. These models learn low-complexity binding functions characterized by multiplicative interactions between concepts, enabling systematic generalization. Code is publicly available at GitHub.

## 1. Introduction

Given a picture of a scene with two objects of different colors and shapes, humans can easily determine which color is associated with which shape, an ability often referred to as concept binding (Treisman & Gelade, 1980; Treisman, 1998). While straightforward for humans, this has proved challenging for modern vision-language embedding systems such as CLIP (Radford et al., 2021). These models often behave like bag-of-concepts systems: they reliably recognize

individual concepts (e.g., *red*, *square*), yet struggle to correctly represent their combinations in multi-object scenes (e.g., distinguishing *red square* vs. *blue square*) (Yuksekgonul et al., 2023; Lewis et al., 2024; Tang et al., 2023). At the same time, within individual modalities of vision and language, object-related information can often be recovered from embeddings using probes (Koishigarina et al., 2026). In other words, despite their failures in multi-object settings, these models appear to internally encode object information to some extent. This tension motivates a closer examination of the geometry of CLIP's binding: if object information is present, how is it represented, and why does it fail to align across modalities?

We begin by analyzing how multi-object scenes are represented in CLIP. We find that scene embeddings exhibit a clear additive structure: they decompose into sums of object representations. This structure explains why uni-modal probes can recover object information—object-level components are explicitly present in the embedding space and can be manipulated directly. However, this geometric structure alone does not explain binding. We show that CLIP's binding function, which maps concepts to object representations, is high-complexity in the sense that it cannot be captured by a simple MLP mapping. Instead, the mapping from concepts to objects differs across concept combinations, effectively requiring memorization of object identities. This prevents generalization to unseen objects and makes it difficult for the image and text encoders to learn a shared binding mechanism, despite exhibiting uni-modal binding.

We then ask whether this limitation is fundamental. We show that it is not. In controlled transformer-based dual-encoder models trained from scratch on synthetic multi-object data, binding generalization emerges with sufficient data coverage. These models learn low-complexity binding functions that reuse structure across concept combinations and generalize to unseen objects. Finally, we analyze the structure of these generalizing binding functions. We find that models that generalize implement multiplicative interactions between concepts, rather than purely additive composition. This multiplicative structure enables generalization and explains how different modalities can produce aligned object representations for unseen concept combinations.

Together, these results show that binding failures in CLIP arise not from the absence of object structure, but from the

---

[*]Equal contribution. [1]University of Tübingen, Tübingen AI Center [2]KAIST AI. Email to: <arnas.uselis, darina.koishigarina@uni-tuebingen.de>.

*Proceedings of the 43rd International Conference on Machine Learning*, Seoul, South Korea. PMLR 306, 2026. Copyright 2026 by the author(s).

complexity of the binding function it learns. Embedding models are capable of generalizable binding, but doing so requires learning a low-complexity, systematic concept-to-object mapping that can be shared across modalities.

## 2. Related work

**Binding.** The binding problem asks how models associate features that belong to the same object (Greff et al., 2020; Feng & Steinhardt, 2024). Surveys and empirical studies examine binding limits and emergent symbolic mechanisms (Campbell et al., 2024; Assouel et al., 2026; Jeong et al., 2026). In autoregressive language models, mechanistic studies have identified position-independent binding IDs that link related tokens (Feng & Steinhardt, 2024; Feng et al., 2025). Binding at patch-level of DINO and CLIP has also been studied (Li et al., 2026), showing that vision models naturally develop an ability to bind at a patch level. However, such mechanisms operate at the token level and it remains unclear how embedding models that produce a single vector per input could implement analogous structure. In contrast, we study how binding is realized in the geometry of embedding models: we ask how object representations are constructed from concepts, and argue that the complexity of this mapping explains cross-modal binding failures.

**Binding in embedding models.** CLIP has been observed to exhibit uni-modal binding (Koishigarina et al., 2026) but fail cross-modally (Lewis et al., 2024; Tang et al., 2023). These failures have been attributed to encoder-level weaknesses such as difficulty with fine-grained details, negations, and spatial reasoning (Tong et al., 2024b;a; Kamath et al., 2023; Li et al., 2025) and over-reliance on concept-level information (Huang et al., 2025). Several works propose fine-tuning approaches to improve CLIP's compositional understanding, including word order sensitivity and attribute-object association (Yuksekgonul et al., 2023; Ma et al., 2023; Hsieh et al., 2024) and argue of the importance of data (Gurung et al., 2025). Some works suggest that concept recognition and object recognition (binding) are fundamentally at odds, suggesting a trade-off (Kang et al., 2025). In contrast, we show that both concept and object recognition can coexist in the same embedding space, explain how uni-modal binding can succeed via linear probes, and argue that cross-modal binding failures stem from the complexity of how concepts are combined into objects, not from an inherent incompatibility.

**Compositional generalization.** Research on compositional generalization investigates how models can generalize to novel combinations of known parts. Many approaches attempt to inject this ability into neural networks (Wang et al., 2024; Mahajan et al., 2025; Jarvis et al., 2024), while others study conditions under which it can be achieved (Lippl & Stachenfeld, 2025; Montero et al., 2021; Dittadi et al.,

2021; Liang et al., 2025; Schott et al., 2022; Kempf et al., 2025; Okawa et al., 2023; Uselis et al., 2026). However, this literature largely focuses on single-object settings with factorized concept spaces. For CLIP specifically, generalization of concept recognition to novel combinations has been studied (Uselis et al., 2025), but generalizing binding, object-level recognition, to novel combinations remains difficult (Lewis et al., 2024). In contrast, we study multi-object scenes and find that generalization of binding is possible and emerges with data scale: at small scales, object recognition fails to generalize (consistent with prior findings), but concept recognition of novel objects remains robust; at larger scales, both concept and object recognition generalize.

**Geometry of learned representations.** Many works study the shape of learned features. General notions of the Linear Representation Hypothesis mostly report linearity in language models (Jiang et al., 2024; Park et al., 2023). Abbasi et al. (2024) find evidence of disentanglement in CLIP models, Trager et al. (2023) show additive decomposition in CLIP text encoder, and Uselis et al. (2025); Berasi et al. (2025) show it in vision encoder. However, these analyses typically characterize geometry in terms of single objects. In contrast, we extend this to multi-object scenes and show that scene embeddings admit a two-level additive decomposition into objects and concepts.

## 3. Formalisation of binding

Before analyzing how embedding models represent multi-object scenes, we formalize the problem. We define what it means to recognize concepts, recognize objects, and bind them together. This formalization guides §4 and §5.

We illustrate the setup in Fig. 1. We assume images and captions depict scenes. A scene contains one or more objects. Each object is specified by values of a fixed set of concepts (e.g. color and shape). We follow prior work (Okawa et al., 2023) in defining objects as combinations of concept values (Fig. 1(a)) and extend this formalisation to scenes containing multiple objects (Fig. 1(b)).

**Definition 3.1** (Concept space). Let the $C$ concepts be indexed by $i \in \{1, \ldots, C\}$, where concept $i$ takes values in a set $\mathcal{C}_i$ (often $|\mathcal{C}_i| = V$ for all $i$, where $V$ is the number of values per concept). The *concept space* is the Cartesian product $\mathcal{C} = \mathcal{C}_1 \times \cdots \times \mathcal{C}_C$.

An *object* is a vector $\boldsymbol{o} = (c_1, \ldots, c_C) \in \mathcal{C}$, where $c_i \in \mathcal{C}_i$ specifies the value for concept $i$. An object therefore corresponds to a particular combination of concept values.

**Definition 3.2** (Scene space). Let $O_{\max} \in \mathbb{N}$ denote the maximum number of objects in a scene. For each $m \in \{1, \ldots, O_{\max}\}$, let $\mathcal{C}^m$ be the set of ordered $m$-tuples of objects.

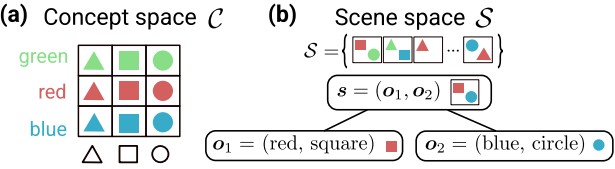

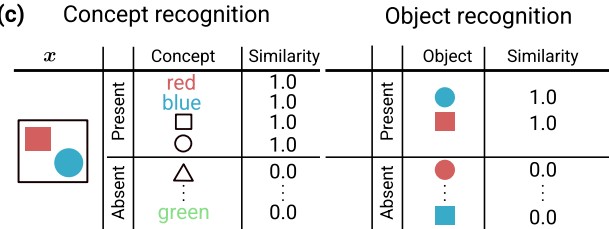

*Figure 1.* **Schematic of the binding setup. (a)** Example concept space $\mathcal{C}$ (e.g., color $\times$ shape), where each object is a tuple of concept values. **(b)** Example scene space $\mathcal{S}$, where a scene $\boldsymbol{s}$ is a tuple of objects (here, two objects $(\boldsymbol{o_1}, \boldsymbol{o_2})$). **(c)** Example of the two recognition criteria: *concept recognition* ranks present concept values above absent ones (Def. 3.3), while *object recognition* ranks present objects above absent concept combinations (Def. 3.4).

The *scene space* is

$$\mathcal{S} = \bigcup_{m=1}^{O_{\max}} \mathcal{C}^m$$

A *scene* is a tuple $\boldsymbol{s} = (\boldsymbol{o_1}, \boldsymbol{o_2}, ..., \boldsymbol{o_m}) \in \mathcal{S}$, describing a collection of $m$ objects. We assume that each scene $\boldsymbol{s}$ corresponds to some datapoint $\boldsymbol{x_s} \in \mathcal{X}$ (e.g. an image).

**Models.** We study embedding models that encode inputs and queries independently. An input corresponds to a scene, such as an image, while a query specifies the information to retrieve, such as a concept value or an object (e.g. CLIP).

We denote the input encoder by $f : \mathcal{X} \to \mathbb{R}^d$ and the query encoder by $q : \mathcal{Y} \to \mathbb{R}^d$. Given an input $\boldsymbol{x_s}$ and a query $\boldsymbol{y} \in \mathcal{Y}$, the matching score is given by cosine similarity, $s(\boldsymbol{x_s}, \boldsymbol{y}) = \frac{f(\boldsymbol{x_s})^\top q(\boldsymbol{y})}{\|f(\boldsymbol{x_s})\| \|q(\boldsymbol{y})\|}$. We refer to $(f, q)$ as a model.

For a scene $\boldsymbol{s} = (\mathbf{o}^{(1)}, \ldots, \mathbf{o}^{(m)}) \in \mathcal{S}$, we define the set of values taken by concept $i$ across all objects in the scene as $V_i(\boldsymbol{s}) := \{c_i^{(1)}, \ldots, c_i^{(m)}\} \subseteq \mathcal{C}_i$. We also define the set of objects present in the scene as $O(\boldsymbol{s}) := \{\mathbf{o}^{(1)}, \ldots, \mathbf{o}^{(m)}\} \subseteq \mathcal{C}$.

**Binding ability.** A model exhibits binding ability when it has (Fig. 1(c)): (1) *concept-level recognition*: recognize which concept values are present in a scene, (2) *object-level recognition*: recognize which objects are present in a scene.

**Definition 3.3** (Concept recognition). A model $(f, q)$ recognizes concepts if for any scene $\boldsymbol{s} \in \mathcal{S}$ and any concept $i \in [C]$, the model assigns higher scores to all concept values present in the scene than to any absent ones:

$$\min_{v \in V_i(\boldsymbol{s})} s(\boldsymbol{x_s}, \boldsymbol{y}_{i,v}) > \max_{v' \in \mathcal{C}_i \setminus V_i(\boldsymbol{s})} s(\boldsymbol{x_s}, \boldsymbol{y}_{i,v'}). \quad (1)$$

While concept recognition captures the presence of individual concept values, it does not test whether the model can identify which values belong together. This distinction is formalized by object recognition.

**Definition 3.4** (Object recognition). A model $(f, q)$ recognizes objects if for any scene $\boldsymbol{s} \in \mathcal{S}$, the model assigns higher scores to all objects present in the scene than to any objects not present:

$$\min_{\mathbf{o} \in O(\boldsymbol{s})} s(\boldsymbol{x_s}, \boldsymbol{y_o}) > \max_{\mathbf{o}' \in \mathcal{C} \setminus O(\boldsymbol{s})} s(\boldsymbol{x_s}, \boldsymbol{y_{o'}}). \quad (2)$$

Concept recognition and object recognition capture complementary aspects of how a model represents scenes. Binding concerns the ability to support both levels consistently.

**Definition 3.5** (Binding). A model exhibits binding if it satisfies both *concept recognition* (Def. 3.3) and *object recognition* (Def. 3.4), i.e., it can identify which concept values are present in a scene and which values belong together.

The definitions above characterize binding as a behavioral property. We distinguish *cross-modal binding*, where the model $(f, q)$ directly satisfies concept and object recognition via cosine similarity, from *uni-modal binding*, where a probe must be trained on top of embeddings within a single modality. Prior work has shown that CLIP exhibits uni-modal binding but fails cross-modally (Koishigarina et al., 2026). To understand why, we examine how embeddings are constructed from scenes.

**Definition 3.6** (Binding function). For a dual-encoder model $(f, q)$, we define the *binding functions* $B_{\text{img}} : \mathcal{S} \to \mathbb{R}^d$ and $B_{\text{txt}} : \mathcal{S} \to \mathbb{R}^d$ as

$$B_{\text{img}}(\boldsymbol{s}) := f(\boldsymbol{x_s}), \qquad B_{\text{txt}}(\boldsymbol{s}) := q(\boldsymbol{y_s}), \quad (3)$$

mapping scenes to embeddings within each modality.

Intuitively, cross-modal binding requires $B_{\text{img}}$ and $B_{\text{txt}}$ to produce compatible embeddings for the same scene, including novel ones. By Occam's razor, among functions consistent with the training scenes, those with simpler, more reusable structure are more likely to generalize identically off-domain (Elmoznino et al., 2025): if both binding functions admit a low-complexity, compositional rule, fewer mappings are consistent with the observations, so the two encoders are more likely to converge on the same rule and remain aligned on unseen objects. If they are high-complexity, each modality may instead fit the training domain in a combination-specific way and disagree off-domain, preventing cross-modal alignment.

In the following sections, we first study the *geometry* of CLIP's binding functions: how scene embeddings decompose into objects and concepts (§4). We then investigate whether these functions can be approximated using concept indices and study controlled models in which binding generalizes (§5).

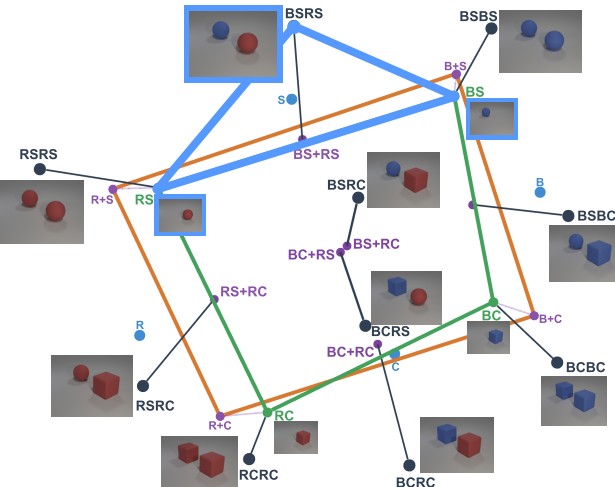

*Figure 2.* **Additive structure in two-object scene embeddings.** MDS projection of CLIP embeddings for single- and two-object scenes that vary in color and shape (distances in the plot approximate embedding distances). Labels use $R/B$ for red/blue and $C/S$ for cube/sphere; points correspond to embeddings of the associated single concepts, single-object scenes, and their two-object combinations. Two-object scene embeddings lie near the line segments between the corresponding single-object embeddings; the highlighted triplet (blue outline) illustrates this geometry for one pair of single-object scenes and their combined two-object scene.

## 4. Geometry of embedding models's binding

CLIP embeddings have been shown to encode binding information within each modality, while this information is misaligned across modalities (Koishigarina et al., 2026). However, it remains unclear how this binding information is represented in the embedding space and how it becomes accessible to probes. To address this, we examine the geometry of CLIP embeddings and find that scene embeddings decompose into object components, which further decompose into concept components (§4.1). We show that these object components are directly responsible for object recognition (§4.2), explaining how binding information can be recovered uni-modally. While prior work mostly focused on single-object representations (Trager et al., 2023; **?**; Berasi et al., 2025), we study embeddings in the multi-object setting. Metric definitions are provided in §C.2. Descriptions of the visual datasets and text compositions are given in §C.1 and §C.3.

### 4.1. Embeddings decompose into objects and concepts

To gain intuition about the geometry of CLIP embeddings, we visualize them in Fig. 2. It shows an MDS (Borg & Groenen, 2005) projection of embeddings for two-object scenes. Two-object scene embeddings lie roughly between the embeddings of the corresponding single-object scenes, suggesting that scene embeddings are approximately sums of their constituent object embeddings. Motivated by this geometric structure, we study how multi-object scenes are

*Table 1.* **Estimators for object embeddings $u_o$ and concept embeddings $u_c$.** Each method averages the image encoder $f$ over a subset of scenes $\mathcal{S}$ containing the target object or concept value. The AVG+POS variant conditions on the position $k$ of the object in the scene tuple $s = (o_1, \ldots, o_m)$.

| Method | Subset | Estimator |
|---|---|---|
| AVG | $\mathcal{D}_o = \{s \in \mathcal{S} : o \in O(s)\}$ | $u_o^{(\mathrm{avg})} = \mathbb{E}_{s \sim \mathcal{D}_o}[f(x_s)]$ |
| AVG+POS | $\mathcal{D}_o^{(k)} = \{(o_1, \ldots, o_m) \in \mathcal{S} : o_k = o\}$ | $u_{o,k}^{(\mathrm{avg+pos})} = \mathbb{E}_{s \sim \mathcal{D}_o^{(k)}}[f(x_s)]$ |
| SINGLE-OBJ | $\mathcal{D}_o^{(1)} = \{s \in \mathcal{S} : s = (o)\}$ | $u_o^{(\mathrm{single\text{-}obj})} = \mathbb{E}_{s \sim \mathcal{D}_o^{(1)}}[f(x_s)]$ |
| CONCEPT | $\mathcal{D}_c = \{s \in \mathcal{S} : c \in V_i(s)\}$ | $u_c = \mathbb{E}_{s \sim \mathcal{D}_c}[f(x_s)]$ |

represented in CLIP. We consider scenes with two objects. Throughout the section, we will let $s = (o_1, o_2)$ denote a scene and $x_s$ denote the corresponding input.

We hypothesize that the embedding space exhibits a hierarchical additive structure, where scenes decompose into objects and objects further decompose into concepts:

1. *(Level-I)* A scene embedding approximately decomposes into a sum of object embeddings.
2. *(Level-II)* An object embedding approximately decomposes into a sum of concept embeddings.

Under this hypothesis, the embedding of a two-object scene can be written as follows. Writing each object as a tuple of concept values, $o_1 = (c_{1,1}, \ldots, c_{1,C})$ and $o_2 = (c_{2,1}, \ldots, c_{2,C})$:

$$f(x_s) \approx u_{o_1} + u_{o_2} \approx \sum_{i=1}^{C} u_{c_{1,i}} + \sum_{i=1}^{C} u_{c_{2,i}}. \qquad (4)$$

We estimate object and concept embeddings using the methods summarized in Tab. 1. We empirically evaluate both levels of this decomposition below.

**Scenes decompose into object components (*Level-I*).** We test the first claim of the Level-I decomposition: that scene embeddings decompose into object-level components. We estimate object embeddings $u_o$ according to Tab. 1. We then reconstruct scene embeddings by summing the corresponding object components. We evaluate the reconstruction using the retrieval score, probing accuracy, and $R^2$.

Tab. 2 shows that CLIP scene embeddings are well approximated by sums of object components (text: $R^2 = 0.90/0.92$; image PUG:SPARE CLIP: $R^2 = 0.75/0.84$), supporting the proposed object-level decomposition. The same decomposition also extends to 3-object CLEVR scenes (including with occlusions; §D.3) and to natural images generated by Gemini Nano Banana 2 (§D.4).

**Editing scene embeddings via object replacement.** The Level-I decomposition in (4) has a concrete implication. If a scene embedding is well approximated by a sum of object components, then editing a multi-object embedding by removing one object component and inserting another should produce a meaningful counterfactual representation.

*Table 2.* **Object-based reconstruction preserves most of the scene variance and retains predictive structure.** We report $R^2$, retrieval accuracy, and probing accuracy across models. Entries for $R^2$ are AVG / AVG+POS; retrieval and probing are AVG+POS only.

| Dataset | Model | $R^2$ | Retrieval | Probing |
|---|---|---|---|---|
| Text | Guess | — | 1 / 160K | 1 / 400 |
| | Random | 0.47 / 0.69 | 0.42 | 0.82 |
| | CLIP | 0.90 / 0.92 | 0.97 | 0.99 |
| PUG:SPARE | Random | 0.47 / 0.53 | 0.25 | 0.90 |
| | CLIP | 0.75 / 0.84 | 0.93 | 0.98 |
| | DINOv2 | 0.78 / 0.86 | 0.86 | 0.98 |
| CLEVR-2D | CLIP | 0.75 / 0.77 | 0.82 | 0.99 |
| | DINOv2 | 0.78 / 0.79 | 0.68 | 0.97 |
| CLEVR | CLIP | 0.78 / 0.83 | 0.94 | 0.96 |
| | DINOv2 | 0.85 / 0.88 | 0.86 | 0.96 |

*Table 3.* **Object-level decomposition enables controlled edits of scene embeddings.** Probing and retrieval performance under object replacement interventions. Each entry corresponds to AVG, AVG+POS, and SINGLE-OBJ object embeddings.

| Dataset | Model | Probing | Retrieval |
|---|---|---|---|
| Text | CLIP | *0.99 / 0.99 / 0.69* | *0.99 / 0.99 / 0.96* |
| CLEVR | CLIP | *0.98 / 0.98 / 0.86* | *1.00 / 1.00 / 0.97* |
| | DINO | *0.97 / 0.98 / 0.77* | *0.89 / 0.95 / 0.86* |
| CLEVR-2D | CLIP | *0.98 / 0.98 / 0.92* | *0.99 / 0.99 / 0.97* |
| | DINO | *0.97 / 0.98 / 0.72* | *0.97 / 0.97 / 0.88* |
| PUG:SPARE | CLIP | *0.94 / 0.95 / –* | *0.86 / 0.94 / –* |
| | DINO | *0.97 / 0.97 / –* | *0.48 / 0.76 / –* |

We examine this through linear operations in embedding space as depicted in Fig. 3.

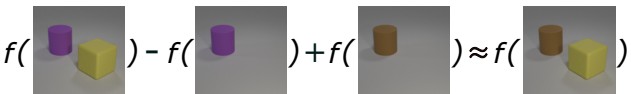

*Figure 3.* **Scene embeddings support object-level editing via linear operations in embedding space.** Subtracting one object embedding and adding another produces a counterfactual embedding corresponding to the edited scene.

**Setup.** Let $s = (o_1, o_2)$ be a two-object scene, and let $o'_1$ be a counterfactual object obtained by changing one concept value of $o_1$ (e.g. color). We construct an edited embedding by substituting the object component of $o_1$ with that of $o'_1$:

$$\tilde{z} := f(x_s) - u_{o_1} + u_{o'_1}. \qquad (5)$$

We evaluate edited embeddings $\tilde{z}$ with retrieval and probing metrics; full details are in §D.2. We report results for text encoders (§C.3) and for image encoders on CLEVR, CLEVR-2D, and PUG:SPARE (§C.1).

**Results.** As shown in Tab. 3, replacing an object in a multi-object scene embedding with another object produces embeddings that behave like the intended counterfactual scenes under both probing and retrieval (e.g., CLIP on CLEVR

achieves 1.00 retrieval with scene-averaged object embeddings). Notably, object embeddings obtained from single-object scenes can be directly used to edit multi-object scene embeddings (e.g., retrieval accuracy 0.97 for CLEVR and CLEVR-2D with CLIP). This supports the Level-I decomposition hypothesis that scene embeddings are composed of additive object-level components.

**Objects decompose into concepts (*Level-II*).** Having established scene-level decomposition, we now analyse whether objects themselves decompose into concepts as in (4). A purely additive concept-level decomposition corresponds to a bag-of-concepts model, which cannot encode within-object associations required for binding. We therefore quantify how much variance is explained by object-only and concept-only components using $R^2$.

*Table 4.* **Concepts explain most scene variance, but objects capture additional structure.** We report $R^2$ for predicting (i) the scene embedding from only objects and (ii) the object and scene embeddings from only concepts, for PUG:SPARE. Values are shown for AVG / AVG+POS object embeddings.

| Model | Scenes from objects $R^2$ | Objects from concepts $R^2$ | Scenes from concepts $R^2$ |
|---|---|---|---|
| Random-embeds | 0.01 | 0.01 | -0.03 |
| Random-init (txt) | *0.47 / 0.69* | *0.56 / 0.80* | *0.39 / 0.56* |
| CLIP-B/32 (txt) | *0.90 / 0.92* | *0.88 / 0.90* | *0.83 / 0.84* |
| Random-init (img) | *0.47 / 0.53* | *0.25 / 0.57* | *0.14 / 0.34* |
| CLIP-B/32 (img) | *0.75 / 0.84* | *0.35 / 0.83* | *0.30 / 0.71* |

**Results.** From Tab. 4, we observe that a bag-of-concepts representation explains a large fraction of embedding variance, up to 84% for the text encoder and 71% for the image encoder. Representing scenes as bags of objects explains roughly 10% more variance. Thus, most variance is captured by concepts, with object representations contributing additional structure beyond a bag-of-concepts.

**Takeaway §4.1:** Scene embeddings decompose into object components, which explain most of the variance and support object edits that remain consistent under probing and retrieval.

## 4.2. Uni-modal binding is explained by object decomposition

In the previous section, we showed that CLIP embeddings decompose into object and concept components. In this section, we demonstrate that these components are directly responsible for the corresponding object and concept recognition in scene embeddings.

**Setup.** We perform targeted component removal on the scene embedding and then fit linear probes on the residuals. Using the decomposition in (4), we define the *concept part* of a scene as $\sum_i u_{c_{1,i}} + \sum_i u_{c_{2,i}}$, and the *object part* as $u_{o_1} + u_{o_2}$. We then form intervened embeddings by sub-

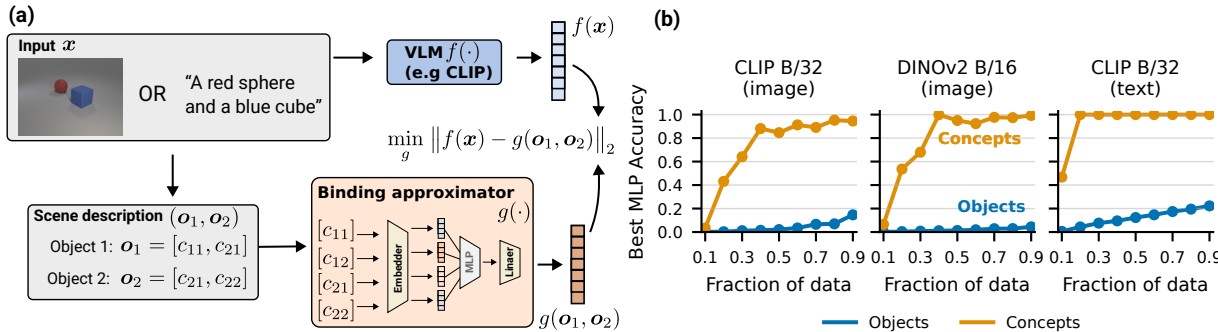

**Figure 4. CLIP's binding function is high-complexity. (a)** We train a binding approximator $g(\boldsymbol{o}_1, \boldsymbol{o}_2)$ (a single-layer MLP) to predict CLIP scene embeddings $f(\boldsymbol{x})$ from concept indices describing the objects present, minimizing (6). **(b)** Maximum accuracy achieved across all MLP capacities and training coverages for concept vs. object recognition on scenes composed of held-out objects. Predicting concepts is high, confirming that additive structure suffices to capture concept-level information. Object recognition is worse, indicating that predicting how concepts combine into objects requires high model complexity.

tracting these components, e.g. $f(\boldsymbol{x}_{\boldsymbol{s}}) - \sum_i \boldsymbol{u}_{c_1,i} - \sum_i \boldsymbol{u}_{c_2,i}$ (remove concepts) or $f(\boldsymbol{x}_{\boldsymbol{s}}) - \boldsymbol{u}_{\boldsymbol{o}_1} - \boldsymbol{u}_{\boldsymbol{o}_2}$ (remove objects), and train probes on the resulting residual embeddings. As a control, we repeat the same removals using *permuted-ID* concept/object components (details in §D.1).

**Results.** Tab. 5 demonstrates that concept recognition and object recognition in scenes depend on their corresponding components in the scene embeddings. For CLIP-B/32, subtracting concept components nearly eliminates concept decoding (text: $1.00 \rightarrow 0.06$, image: $0.94 \rightarrow 0.05$) while largely preserving object decoding (text: $1.00 \rightarrow 0.99$, image: $0.96 \rightarrow 0.85$). In contrast, subtracting object components collapses both object and concept decoding (text: 0.04/0.05, image: 0.01/0.02). Permuted-ID controls do not produce comparable degradation. Removing object components that are not present in the scene has little effect on decoding accuracy, confirming that the performance drop is specific to the correct object component.

These results explain why prior work observes binding in CLIP embeddings. Object components in a scene embedding encode concept information jointly, allowing probes to recover the correct combinations of concepts.

> **Takeaway §4.2:** Object components in scene embeddings are responsible for object recognition, which explains why the binding is observed uni-modally via probes.

## 5. Binding generalization

Having established that scene embeddings decompose into objects, explaining uni-modal binding, we next examine why cross-modal binding fails through the complexity and generalization of the underlying binding functions.

### 5.1. CLIP's binding function does not generalize

Inspired by prequential/MDL perspectives that interpret simplicity as fast generalization from limited data (Blier & Ollivier, 2018; Elmoznino et al., 2025), we measure how

*Table 5.* **Removing object or concept components reduces recognition accuracy for the corresponding prediction task.** We report probe accuracy after subtracting components from the embedding (remove concepts/objects) or their permuted-ID controls, for both text and image encoders and their random-init counterparts.

| Model | Intervention | Text | | Image | |
|---|---|---|---|---|---|
| | | Conc. | Obj. | Conc. | Obj. |
| Random | None | 0.94 | 0.60 | 0.76 | 0.71 |
| | Subtract concepts | 0.02 | 0.26 | 0.06 | 0.66 |
| | Subtract objects | 0.01 | 0.01 | 0.04 | 0.00 |
| | Permute concepts | 0.87 | 0.68 | 0.74 | 0.72 |
| | Permute objects | 0.83 | 0.93 | 0.78 | 0.88 |
| CLIP-B/32 | None | 1.00 | 1.00 | 0.94 | 0.96 |
| | Subtract concepts | 0.06 | 0.99 | 0.05 | 0.85 |
| | Subtract objects | 0.05 | 0.04 | 0.02 | 0.01 |
| | Permute concepts | 0.92 | 0.99 | 0.99 | 0.97 |
| | Permute objects | 0.96 | 1.00 | 0.86 | 0.92 |

generalization of a binding function (Def. 3.6) depends on training-set size and MLP capacity (width). Concretely, we study the sample-capacity requirements of a 1-hidden-layer ReLU MLP whose width is varied, trained to approximate a binding map from discrete concept values to scene embeddings. We use MLPs because SGD-trained MLPs favor simple, compressible solutions (Wilson, 2025), so if a simple compositional binding rule exists, we expect this approximator family to find it.

**Setup.** We train an approximator $g : \mathcal{C}^2 \rightarrow \mathbb{R}^d$ to predict scene embeddings from concept indices (Fig. 4(a)). For a scene $\boldsymbol{s} = (\boldsymbol{o}_1, \boldsymbol{o}_2)$ where each object $\boldsymbol{o}_i = (c_{i1}, c_{i2})$ is specified by concept values, we minimize

$$\min_g \sum_{\boldsymbol{s} \in \mathcal{S}_{\text{train}}} \|f(\boldsymbol{x}_{\boldsymbol{s}}) - g(\boldsymbol{o}_1, \boldsymbol{o}_2)\|^2, \qquad (6)$$

where $f(\boldsymbol{x}_{\boldsymbol{s}})$ is the CLIP embedding of scene $\boldsymbol{s}$ and $\mathcal{S}_{\text{train}}$ is the set of training scenes. We vary (1) MLP hidden layer width $\{64, 256, 1024, 4096\}$, and (2) the fraction of objects seen during training $\{10\%, \dots, 90\%\}$. At test time, we evaluate on scenes containing only held-out objects. We mea-

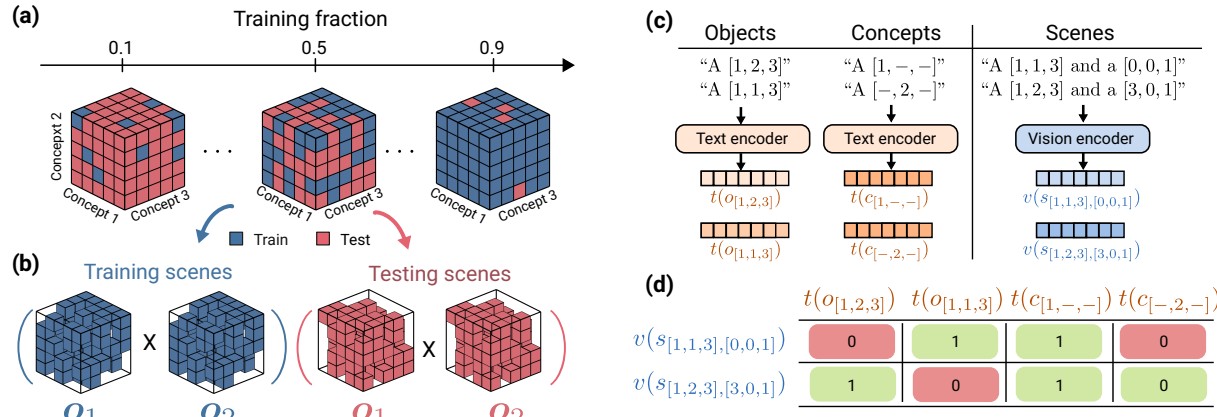

*Figure 5.* **Controlled setup for studying generalizable binding.** We train transformer-based embedding models on synthetic multi-object data to test whether binding can generalize to entirely unseen objects. **(a) Data design:** We vary the training coverage $\rho_{\text{train}}$ from 0.1 to 0.9, controlling what fraction of the object space the model observes during training. **(b) Scene construction:** Training scenes are composed of objects from the training split (blue); test scenes are composed of objects from the held-out split (red). Crucially, test objects have concept configurations that never appeared during training. **(c) Encoder design:** Following CLIP's architecture, we use two independent encoders: a "text" encoder that embeds individual objects or concepts, and a "vision" encoder that embeds full scenes. Both produce embeddings in $\mathbb{R}^{512}$. **(d) Training objective:** We optimize a contrastive retrieval loss using cosine similarity. The table shows a training batch where rows are scene embeddings $v(\mathbf{s})$ and columns are object/concept embeddings $t(\cdot)$; green indicates matching pairs, red indicates mismatches.

sure whether the predicted embeddings support both *concept recognition* (Def. 3.3) and *object recognition* (Def. 3.4) by using the linear probes trained on the scene embeddings. See details in §C.4.

**Results.** Fig. 4**(b)** reports the best accuracy across the MLP configurations. Concept recognition yields $\geq 80\%$ accuracy among all conditons where the fraction of data exceeds 0.3, whereas object recognition is substantially lower, including at large widths and high training coverage. This gap indicates that correctly predicting object-level structure is markedly harder than concept-level structure, suggesting a higher effective complexity of object-level binding under this probe family. Among encoders, the text encoder performs slightly better than the image encoders; CLIP image embeddings outperform DINO but remain far below concept recognition, never exceeding 20% accuracy. The result does not depend on the approximator family: swapping MLPs for XGBoost or Random Forest yields a similar qualitative pattern on CLIP and DINOv2 (§D.6). These results indicate that CLIP's binding function is high-complexity: it resists approximation even by high-capacity MLPs and fails to generalize to novel object combinations.

> **Takeaway §5.1:** For pretrained embedding models, even high-capacity MLPs fail to generalize to unseen concept combinations. This may explain cross-modal binding failures: a complex, non-generalizing mapping prevents encoders from aligning on novel objects.

### 5.2. It is possible to have a generalizable binding model

The previous section showed that CLIP's binding function is hard to approximate and does not generalize to unseen

concept combinations. This raises a natural question: is this failure inherent, or can embedding models learn binding that *does* generalize when the data and training signal are controlled? To answer this, we train CLIP-style dual-encoder transformers on synthetic multi-object scenes where we can precisely vary object-space coverage (Fig. 5).

**Setup.** We illustrate the process in Fig. 5. We use synthetic scenes with up to 2 objects; each object is defined by $C$ concepts with $V$ values ($|O| = V^C$), and we vary $C, V$, and the training fraction of seen objects from 0.1 to 0.9. Train scenes draw only from the training split; test scenes are composed entirely of held-out objects. We use two transformer (Vaswani et al., 2023) encoders (query and scene); we refer to them as "text" and "vision" by analogy only since both process tokenized sequences, differing only in whether they encode object/concept queries or full scenes. Both encoders output embeddings in $\mathbb{R}^{512}$ and have $\approx$20M parameters each. We train end-to-end with AdamW (Loshchilov & Hutter, 2017) on a contrastive retrieval objective (Radford et al., 2021); see §B for full details.

**Metrics.** We evaluate both concept and object recognition. For concepts, we test against all $C \cdot V$ values. For objects, testing against all $V^C$ is infeasible; instead, we use $\mathrm{Swap}(\mathbf{o}_1, \mathbf{o}_2)$, objects formed by exchanging concept values between $\mathbf{o}_1$ and $\mathbf{o}_2$, which directly tests binding by ranking correct object identities against negatives.

**Results.** We observe two key findings (Fig. 6). First, *compositional generalization emerges*: object recognition on held-out objects starts near the bag-of-concepts baseline but rises sharply with coverage, reaching near-perfect accuracy despite all test objects being unseen. Second, *concepts are eas-*

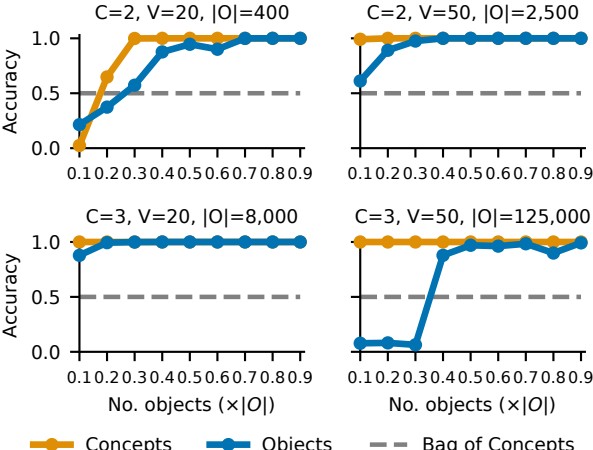

*Figure 6.* **Binding generalization emerges with scale.** Test accuracy on held-out objects as a function of training coverage. Each panel varies object complexity ($C$ concepts, $V$ values, $|O| = V^C$ objects). Concept recognition (orange) generalizes readily; object recognition (blue; binding) requires more coverage but reaches high accuracy, surpassing the bag-of-concepts baseline (dashed).

*ier than objects*: concept recognition generalizes early, while object recognition requires substantially more coverage. When the number of possible objects is small, generalization requires observing more objects, consistent with prior findings in low-combinatorial regimes (Lewis et al., 2024). As the number of possible objects increases, the fraction of objects required for reliable object recognition decreases substantially. For example, when $|O| = 400$, high accuracy is achieved only after observing roughly 50% of objects, whereas for $|O| \geq 2{,}500$, generalization emerges at around 30% coverage. In the largest setting ($|O| = 125{,}000$), increasing coverage from 30% to 40% yields a sharp transition from below-chance to near-perfect object recognition. Notably, this occurs despite the corresponding scene space being extremely large (e.g., $(50^3)^2 \approx 15$ billion two-object scenes for $C = 3, V = 50$).

> **Takeaway §5.2:** When models are trained from scratch on sufficient data, binding *does* generalize: object recognition (binding) can approach perfect accuracy.

## 5.3. Binding functions of generalizing models tend to have low complexity functions

In §5.1 we operationalized *binding complexity* via generalization of an MLP approximator: even wide MLPs struggled to predict CLIP embeddings on held-out objects, revealing a high-complexity, non-generalizing binding map. Here we apply the same diagnostic to the controlled models from §5.2 and test the converse prediction: models that *do* generalize binding should admit a low-complexity (easy-to-approximate) concept-to-object mapping.

**Setup.** We measure binding complexity exactly as in §5.1: we train MLPs of varying width to predict scene embed-

dings from discrete concept indices and evaluate on scenes composed of held-out objects, reporting concept vs. object recognition on the predicted embeddings. If binding is low-complexity, small MLPs should generalize; if it is high-complexity (as in CLIP), generalization should require large capacity and still fail. We further test this in the visual domain by training scene encoders from scratch on pixel inputs, including noisy and occluded settings (see §E).

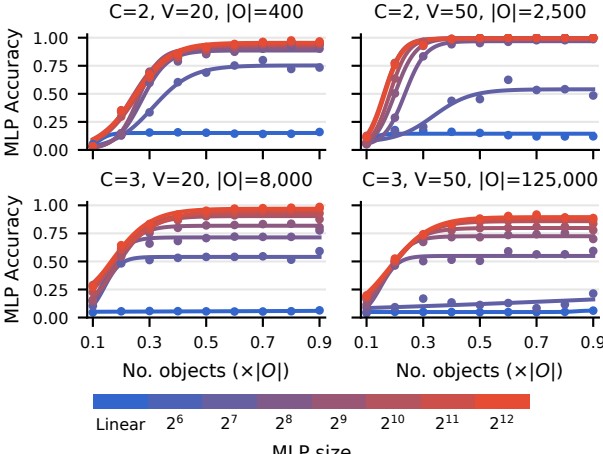

*Figure 7.* **Binding functions can be approximated by low-capacity models.** Even single-layer MLPs with small hidden dimensions achieve high accuracy, suggesting that the learned binding operation has low computational complexity. Each panel varies the number of concepts $C$ and concept values $V$.

**Results.** Fig. 7 shows that for controlled models that generalize, the binding map is easy to approximate in the sense that even small nonlinear MLPs achieve strong accuracy. At the same time, as the object space grows (larger $C$ and $V$), the performance gap between the smallest nonlinear MLP and the largest-capacity MLP increases, showing that the binding map does become moderately more complex with scale. These findings are analogous when the scene encoder is trained from scratch directly on pixel inputs (§E). This contrasts with §5.1, where even high-capacity MLPs fail to generalize object recognition on CLIP embeddings.

> **Takeaway §5.3:** In generalizing transformer-based models, binding function tends to be low-complexity: even small nonlinear MLPs generalize well.

## 5.4. Multiplicative structure best explains binding

We next ask what functional form the learned binding takes. We compare simple, structured probes that predict scene embeddings from concept values via additive and multiplicative composition. The Additive probe is the bag-of-concepts baseline. The Per-obj. products and Global product probes introduce the simplest possible departure from additivity: product interactions between concept embeddings, which can produce a unique representation for each object combination. In this subsection, we consider scenes of two ordered

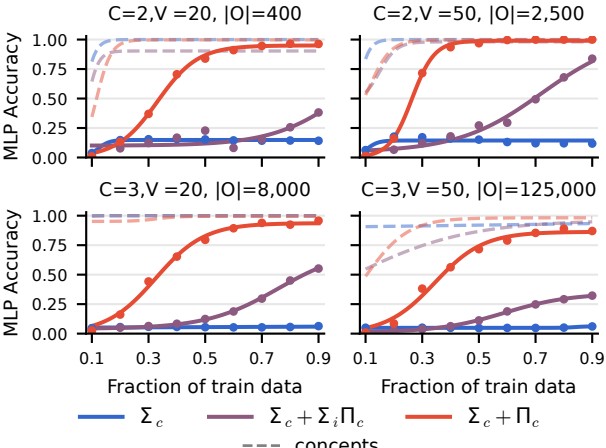

*Figure 8.* **Multiplicative structure best explains binding.** Accuracy of the Additive (—), Per-obj. products (—), and Global product (—) probes for predicting scene embeddings from concept indices, evaluated on held-out objects. Dashed: concept recognition; solid: object recognition.

objects, $\boldsymbol{s} = (\boldsymbol{o}_1, \boldsymbol{o}_2)$, where each object is described by two concept types (e.g., color and shape), so $\boldsymbol{o}_i = (c_{i1}, c_{i2})$ with $c_{ik} \in \mathcal{C}_k$ the value of concept type $k$ in the $i$-th object. The parameters we train are concept-value embeddings: $\boldsymbol{u}_{k,c}$ is indexed by concept type $k$ and value $c \in \mathcal{C}_k$, shared across the two object positions. The multiplicative embedding $\boldsymbol{v}_{i,k,c}$ carries an additional index $i \in \{1, 2\}$ for the object's position, so its products in Per-obj. products (within an object) and Global product (across both objects) yield a distinct vector for every *combination* of concept values — the binding signal that pure addition cannot express. Training and testing data are constructed following §C.4.

| | |
|---|---|
| Additive: | $\sum_{i=1}^{2} \sum_{k=1}^{2} \boldsymbol{u}_{k,c_{ik}}$ |
| Per-obj. products: | $\sum_{i=1}^{2} \sum_{k=1}^{2} \boldsymbol{u}_{k,c_{ik}} + \sum_{i=1}^{2} \prod_{k=1}^{2} \boldsymbol{v}_{i,k,c_{ik}}$ |
| Global product: | $\sum_{i=1}^{2} \sum_{k=1}^{2} \boldsymbol{u}_{k,c_{ik}} + \prod_{i=1}^{2} \prod_{k=1}^{2} \boldsymbol{v}_{i,k,c_{ik}}$ |

**Results.** Fig. 8 shows that all probes do well at concept recognition, but the Additive baseline fails at object recognition. Adding multiplicative interactions recovers binding generalization, with the Global product form performing best, especially in larger object spaces.

**Multiplicative structure underlies generalization.** To test whether this relationship holds more broadly than on the four considered models, we examine the correlation between a model's generalization ability and how well its embeddings are approximated by the Global product probe. For each of ~500 models trained with varying hyperparameters across all object space sizes, we fit the Global product probe using 50% of the object space and measure both (1) the model's object recognition accuracy on held-out objects, and (2) the probe's accuracy at predicting scene embeddings. Fig. 9 shows a strong positive correlation: models that generalize well are precisely those whose embeddings are well-approximated by the Global product form. This suggests

that generalizing models are internally implementing something akin to multiplicative binding—a simple functional form that composes concept representations systematically. The correlation weakens slightly for the largest object space ($|O| = 125$k), but the pattern remains clear: the multiplicative form is sufficient to capture how concepts combine, which is why it generalizes to unseen combinations. Applying the same Global product probe to pretrained CLIP and DINOv2 embeddings instead recovers concept recognition but leaves object recognition near zero (§D.7), consistent with CLIP's failure to bind concepts cross-modally.

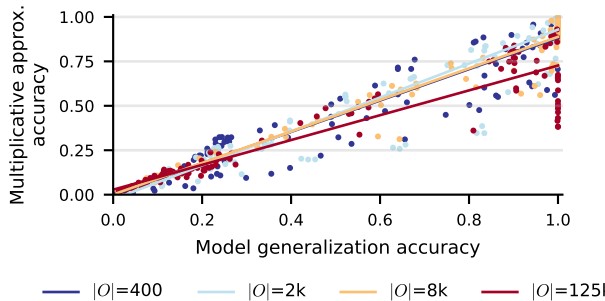

*Figure 9.* **Generalization correlates with multiplicative structure.** Each point is a trained model; x-axis shows object recognition accuracy on scenes composed of held-out objects, y-axis shows how well the Global product probe (trained on 50% of objects) approximates scene embeddings. Colors indicate object space size. Models that generalize also admit simpler (multiplicative) binding.

> **Takeaway §5.4:** Generalizable binding corresponds to multiplicative structure in the embedding space: models whose embeddings admit simple multiplicative composition are precisely those that generalize to unseen object combinations.

## 6. Discussion

In this work we formalized concept binding by distinguishing concept recognition from object recognition, and then analyzed the geometry of multi-object embeddings through this lens. Scene embeddings show a clear additive object-level structure (supporting uni-modal decodability and direct embedding edits), but the concept-to-object mapping appears to be high-complexity and combination-specific, which blocks generalization and makes cross-modal alignment brittle. In controlled transformer dual-encoders trained from scratch, binding generalization emerges with enough data, and the generalizing mapping is best explained by low-complexity structure with multiplicative interactions between concepts.

**Limitations.** Our analysis uses synthetic datasets. To our knowledge, no real-world dataset currently provides the controlled, combinatorially complete structure required to study binding in our setting. In addition, our notion of complexity is necessarily relative to a chosen approximator family, since Kolmogorov complexity is not computable in practice.

## Acknowledgments

The authors thank Declan Campbell for helpful discussions and comments on initial drafts of this work, Alexander Rubinstein for insightful discussions, and the anonymous reviewers for their valuable feedback. This work was supported by the Tübingen AI Center. Arnas Uselis and Darina Koishigarina were supported by the International Max Planck Research School for Intelligent Systems (IMPRS-IS). Seong Joon Oh was supported by the Institute for Information & Communications Technology Planning & Evaluation (IITP) grant funded by the Korea government (MSIT) (RS-2019-II190075, Artificial Intelligence Graduate School Program gram(KAIST)).

## Impact statement

This paper presents work whose goal is to advance the field of machine learning. There are many potential societal consequences of our work, none of which we feel must be specifically highlighted here.

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

# Appendix

# A. Notation summary

*Table 6.* Core notation used in the main text.

| Symbol | Meaning |
|---|---|
| $C$ | Number of concepts per object; $i \in \{1, \dots, C\}$ indexes concepts. |
| $\mathcal{C}_i$ | Set of values for concept $i$; $V$ denotes $|\mathcal{C}_i|$ when uniform. |
| $\mathcal{C}$ | Concept space $\mathcal{C}_1 \times \cdots \times \mathcal{C}_C$. |
| $\mathbf{o} = (c_1, \dots, c_C)$ | Object (concept-value tuple); $c_i \in \mathcal{C}_i$. |
| $O_{\max}$ | Maximum number of objects in a scene. |
| $m$ | Number of objects in a scene. |
| $\mathcal{S}_m$ | Set of ordered $m$-object scenes; $\mathcal{S} = \bigcup_{m=1}^{O_{\max}} \mathcal{S}_m$. |
| $\mathbf{s}$ | Scene (tuple of objects). |
| $\boldsymbol{x}_\mathbf{s}$ | Datapoint (e.g., image) depicting scene $\mathbf{s}$. |
| $\mathcal{X}, \mathcal{Y}$ | Input space and query space. |
| $f, g$ | Input encoder $f : \mathcal{X} \to \mathbb{R}^d$ and query encoder $g : \mathcal{Y} \to \mathbb{R}^d$. |
| $d$ | Embedding dimension. |
| $s(\boldsymbol{x}_\mathbf{s}, \boldsymbol{y})$ | Cosine similarity between scene and query embeddings. |
| $\boldsymbol{y}_{i,v}, \boldsymbol{y}_\mathbf{o}$ | Query for concept $i$ value $v$; query for object $\mathbf{o}$. |
| $V_i(\mathbf{s})$ | Set of concept-$i$ values present in scene $\mathbf{s}$. |
| $O(\mathbf{s})$ | Set of objects present in scene $\mathbf{s}$. |
| $\boldsymbol{u}_\mathbf{o}$ | Object embedding; $\boldsymbol{u}_{c_{j,i}}$ is the embedding of the value of concept $i$ for object $\mathbf{o}_j$. |
| $\tilde{\boldsymbol{z}}$ | Counterfactual (edited) scene embedding. |
| $R^2, \boldsymbol{f}$ | Variance explained; $\boldsymbol{f}$ is the mean scene embedding. |
| $\pi_\text{c}, \pi_\text{o}$ | Permutations of concept/object IDs in control ablations. |
| $\boldsymbol{W}, l$ | Linear model weights and input dimension in additive regressions. |
| $g(\mathbf{o}_1, \mathbf{o}_2)$ | Binding-function approximator (MLP) mapping objects to embeddings. |

# B. Details on the controlled models

We train controlled models from scratch on synthetic multi-object scenes represented as token sequences. We use two transformer encoders: one for scenes and one for queries (either a single concept value or an entire object). We optimize a retrieval objective that makes matching scene-query pairs more similar than mismatched pairs (akin to CLIP's contrastive loss (Radford et al., 2021)). Because both scenes and queries are generated procedurally, we can vary the object-space size $V^C$, the maximum number of objects per scene $O_{\max}$, and whether evaluation uses held-out concept combinations. Table 7 summarizes the key parameters used throughout.

*Table 7.* Notation and key configuration parameters for the controlled models.

| Symbol | Meaning |
|---|---|
| $C$ | Number of concepts per object. |
| $V$ | Number of values per concept. |
| $O_{\max}$ | Maximum number of objects per scene. |
| $L_{\max}$ | Maximum sequence length, $L_{\max} = (C + 2)O_{\max} + 1$. |
| $d_{\text{model}}$ | Transformer width. |
| $H$ | Number of attention heads. |
| $L$ | Number of transformer layers. |
| $d_{\text{out}}$ | Output embedding dimension. |
| $K_{\text{concept}}$ | Max number of concept queries per step. |
| $K_{\text{obj}}$ | Max number of object queries per step. |
| $M_{\text{perturb}}$ | Perturbation negatives per positive object. |
| $M_{\text{swap}}$ | Swap negatives generated per scene. |
| $\rho_{\text{train}}$ | Fraction of objects available during training (object-space split). |
| $s_{\text{split}}$ | Seed for the object-space split. |
| `sim` | Similarity function (`cos` or `dot`). |

## B.1. Scene space and tokenization

We parameterize the synthetic environment by the number of concepts $C$, the number of values per concept $V$, and the maximum number of objects per scene $O_{\max}$ (`max_num_objects`). Each object is represented as a $C$-tuple of discrete

concept values $\boldsymbol{v} = (v_1, \ldots, v_C)$ with $v_c \in \{0, \ldots, V-1\}$, and a scene is an ordered list of $n \in \{1, \ldots, O_{\max}\}$ such objects.

**Scene sampling.** We draw the scene length $n \sim \mathrm{Unif}\{1, \ldots, O_{\max}\}$ and then sample $n$ objects independently. By default, each component $v_c$ is drawn uniformly from $\{0, \ldots, V-1\}$. To implement train/test splits, we instead sample objects from a restricted list `allowed_objects`.

**Tokenization.** Table 8 summarizes the sequence format. We map each concept/value pair $(c, v)$ to a token id $t(c, v) = (c-1)V + v$, yielding $CV$ concept-value tokens, and add four extra tokens: `SOO` (start-of-object), `EOO` (end-of-object), `EOS` (end-of-scene), and `PAD`. An object $\boldsymbol{v} = (v_1, \ldots, v_C)$ is encoded as

$$[\texttt{SOO}, \ t(1, v_1), \ldots, t(C, v_C), \ \texttt{EOO}],$$

and a scene is the concatenation of its objects followed by `EOS`, padded to the fixed length

$$L_{\max} = (C+2)O_{\max} + 1.$$

Scenes are generated on the fly by an "infinite" iterator, so training is naturally described in gradient steps rather than epochs.

*Table 8.* Token IDs and sequence formats used for scenes and probe queries.

| Component | Definition |
|---|---|
| Concept-value token | $t(c, v) = (c-1)V + v$ for concept $c \in \{1, \ldots, C\}$, value $v \in \{0, \ldots, V-1\}$. |
| Special tokens | $\texttt{SOO} = CV$, $\texttt{EOO} = CV+1$, $\texttt{EOS} = CV+2$, $\texttt{PAD} = CV+3$. |
| Object sequence | $[\texttt{SOO}, t(1, v_1), \ldots, t(C, v_C), \texttt{EOO}]$. |
| Scene sequence | Concatenate $n$ objects, append `EOS`, pad with `PAD` to length $L_{\max}$. |
| Concept query | $[t(c, v), \texttt{EOS}]$ padded to $L_{\max}$. |
| Object query | $[\texttt{SOO}, t(1, v_1), \ldots, t(C, v_C), \texttt{EOO}, \texttt{EOS}]$ padded to $L_{\max}$. |

**Example.** With $C = 2$ and $V = 10$, the object $\boldsymbol{v} = (3, 7)$ (i.e., $v_1 = 3$, $v_2 = 7$) is encoded as $[\texttt{SOO}, 3, 17, \texttt{EOO}]$ since $t(1, 3) = 3$ and $t(2, 7) = 10 + 7 = 17$.

## B.2. Encoders

Scenes and queries are embedded by two transformer encoders with the same architecture (`SceneEncoder`) but independent parameters: a *scene encoder* $f_{\mathrm{scene}}$ and a *probe encoder* $f_{\mathrm{probe}}$. Each maps a padded length-$L_{\max}$ token sequence to a single embedding via token embeddings, positional encoding, and an $L$-layer Transformer encoder with $H$ attention heads (masking `PAD`). The scene encoder outputs $\boldsymbol{s} \in \mathbb{R}^{d_{\mathrm{out}}}$. The probe encoder outputs $\boldsymbol{q} \in \mathbb{R}^{d_{\mathrm{out}}}$.

For a minibatch of $B$ scenes and $T$ queries, let $\boldsymbol{S} \in \mathbb{R}^{B \times d_{\mathrm{out}}}$ and $\boldsymbol{Q} \in \mathbb{R}^{T \times d_{\mathrm{out}}}$ collect their embeddings row-wise. We form a logit matrix $\boldsymbol{Z} \in \mathbb{R}^{B \times T}$. We then optimze the embeddings $\hat{\boldsymbol{x}} = \boldsymbol{x}/(\|\boldsymbol{x}\|_2 + 10^{-6})$ and define $\hat{\boldsymbol{S}}, \hat{\boldsymbol{Q}}$ by row-wise normalization. We use $\tau = \exp(\tilde{\tau})$ as a learned temperature parameter, akin to CLIP (clamped to $\tau \leq 100$).

## B.3. Queries, labels, and loss

At each training step we sample a batch of scenes and construct a set of query sequences. We form a multi-hot label matrix $\boldsymbol{Y} \in \{0, 1\}^{B \times T}$, where $\boldsymbol{Y}_{ij} = 1$ if query $j$ matches scene $i$. We use two query types:

1. **Concept retrieval.** Concept queries specify a single concept value. We collect the unique concept-value tokens that appear in the batch (excluding `SOO`/`EOO`/`EOS`/`PAD`), subsample up to `num_concept_values_take_max`, and tokenize each as a concept query.

2. **Object retrieval.** Object queries specify a full object. We parse object blocks from the tokenized scenes, take the set of unique objects present in the batch, subsample up to 1024, and tokenize each as an object query. We then augment the query set with hard negatives (random/perturbed objects and attribute swaps).

Concept queries probe whether the model can retrieve scenes based on single attributes, while object queries probe multi-attribute binding.

Given logits $\boldsymbol{Z}$ and labels $\boldsymbol{Y}$, we treat multiple positives by normalizing each row to a target distribution $\boldsymbol{P}_{i\cdot} = \boldsymbol{Y}_{i\cdot}/(\sum_j \boldsymbol{Y}_{ij} + 10^{-10})$ and minimizing the cross-entropy

$$\mathcal{L} = -\frac{1}{B} \sum_{i=1}^{B} \sum_{j=1}^{T} \boldsymbol{P}_{ij} \log \operatorname{softmax}(\boldsymbol{Z}_{i\cdot})_j.$$

Scenes with $\sum_j \boldsymbol{Y}_{ij} = 0$ (none of the sampled queries match) have $\boldsymbol{P}_{i\cdot} = 0$ and contribute zero loss. When both concept and object queries are enabled we sum their losses and backpropagate through both encoders and $\tilde{\tau}$.

### B.4. Optimization and controlled scenarios

Training uses AdamW over the parameters of both encoders and $\tilde{\tau}$, with a linear warmup (first $10\%$ of steps) followed by linear decay (Bergsma et al., 2025). The code can optionally take multiple optimizer steps per sampled minibatch (`num_grad_steps`). Table 9 lists the default hyperparameters and numerical-stability choices.

*Table 9.* Training hyperparameters and numerical-stability choices (defaults unless stated otherwise).

| Component | Setting |
|---|---|
| Training steps | $T = 50{,}000$ gradient steps |
| Batch size | $B = 1024$ scenes per step |
| Optimizer | AdamW over both encoders and $\tilde{\tau}$ |
| Learning rate | $3 \times 10^{-4}$ |
| Weight decay | 0.001 |
| LR schedule | Linear warmup for $0.1T$ steps, then linear decay to 0 |
| Temperature init/clamp | Initialize $\tilde{\tau} = 1.15$, use $\tau = \exp(\tilde{\tau})$ clamped to $\tau \leq 100$ |

# C. Experimental details

## C.1. Dataset details

We use three real-image datasets: PUG:SPARE (Koishigarina et al., 2026) (Fig. 10) and CLEVR (Johnson et al., 2017) (Fig. 11). We additionally introduce CLEVR-2D (Fig. 12), a 2D adaptation of CLEVR in which 3D shapes are replaced by their flat 2D counterparts (cube → square, sphere → circle, cylinder → triangle). Each scene contains two objects; objects vary along two concepts—color (8 values) and shape (3 values)—yielding 576 two-object scenes. Paired captions follow the same template as CLEVR (e.g., *"a purple square and a gray circle"*).

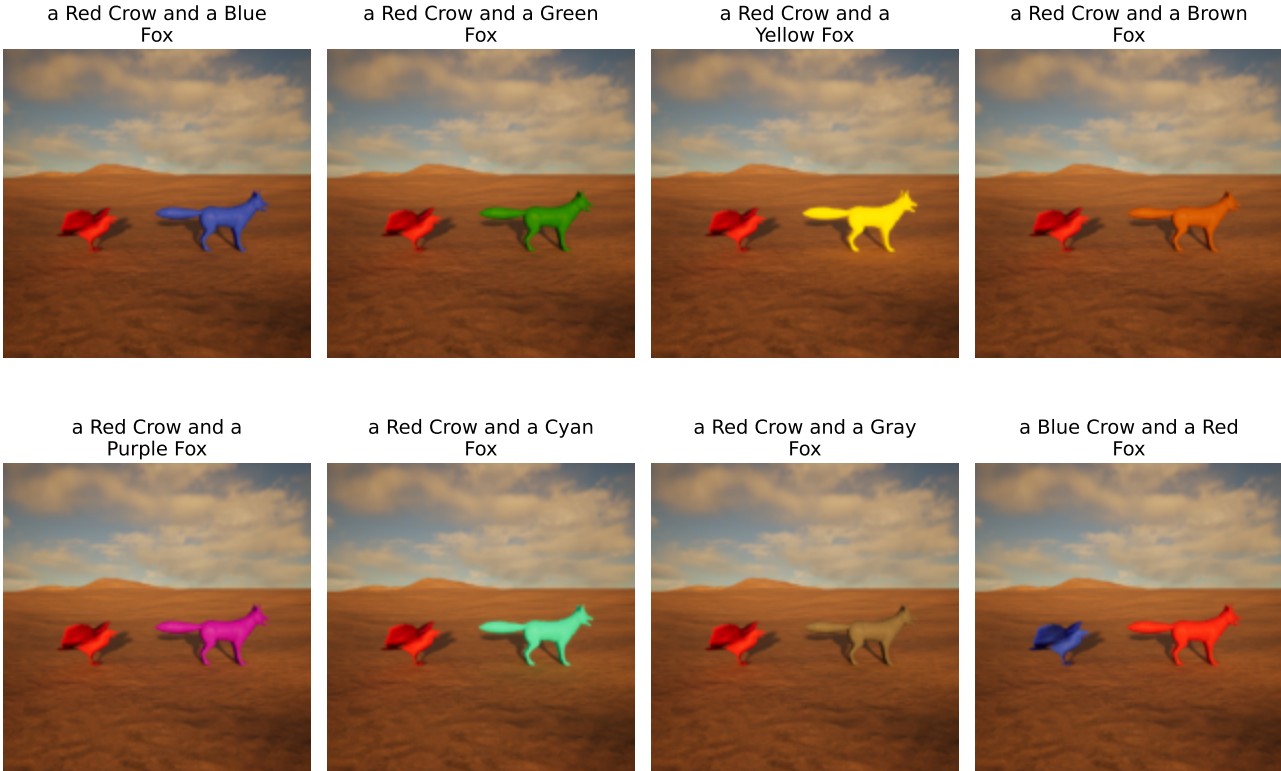

*Figure 10.* **PUG:SPARE dataset samples.** Photorealistic scenes with two objects varying in color and species (12 colors × 8 animals, 7392 scenes total).

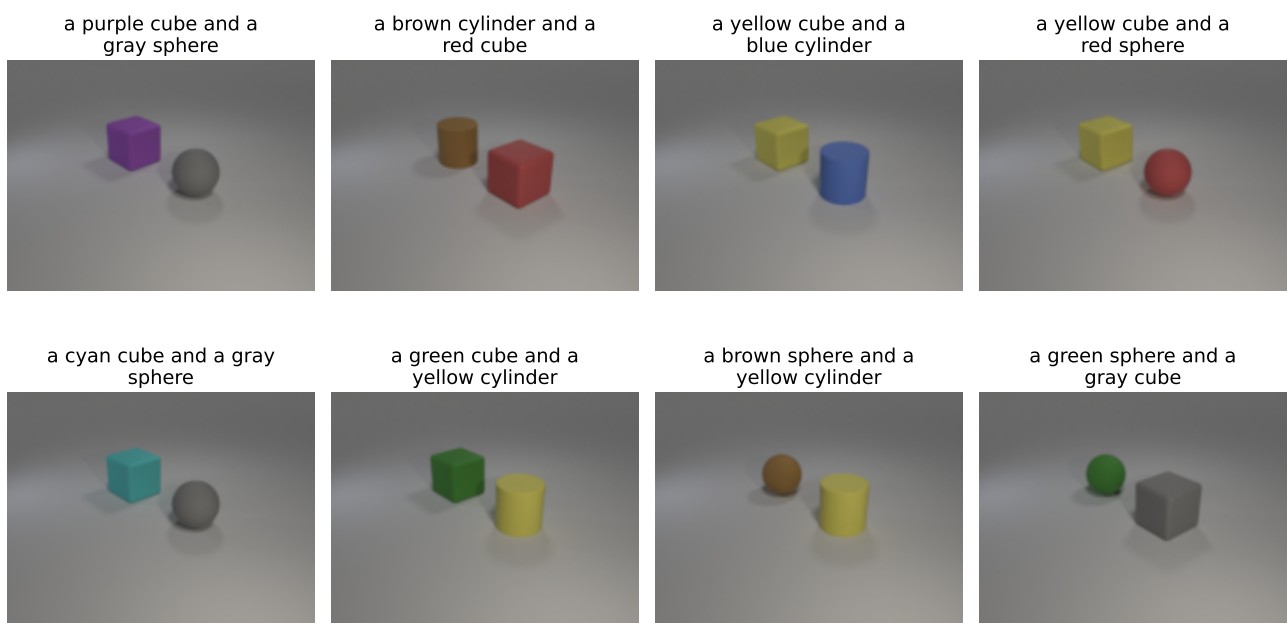

*Figure 11.* **CLEVR dataset samples.** 3D-rendered scenes with two objects varying in color and shape (8 colors × 3 shapes, 576 scenes).

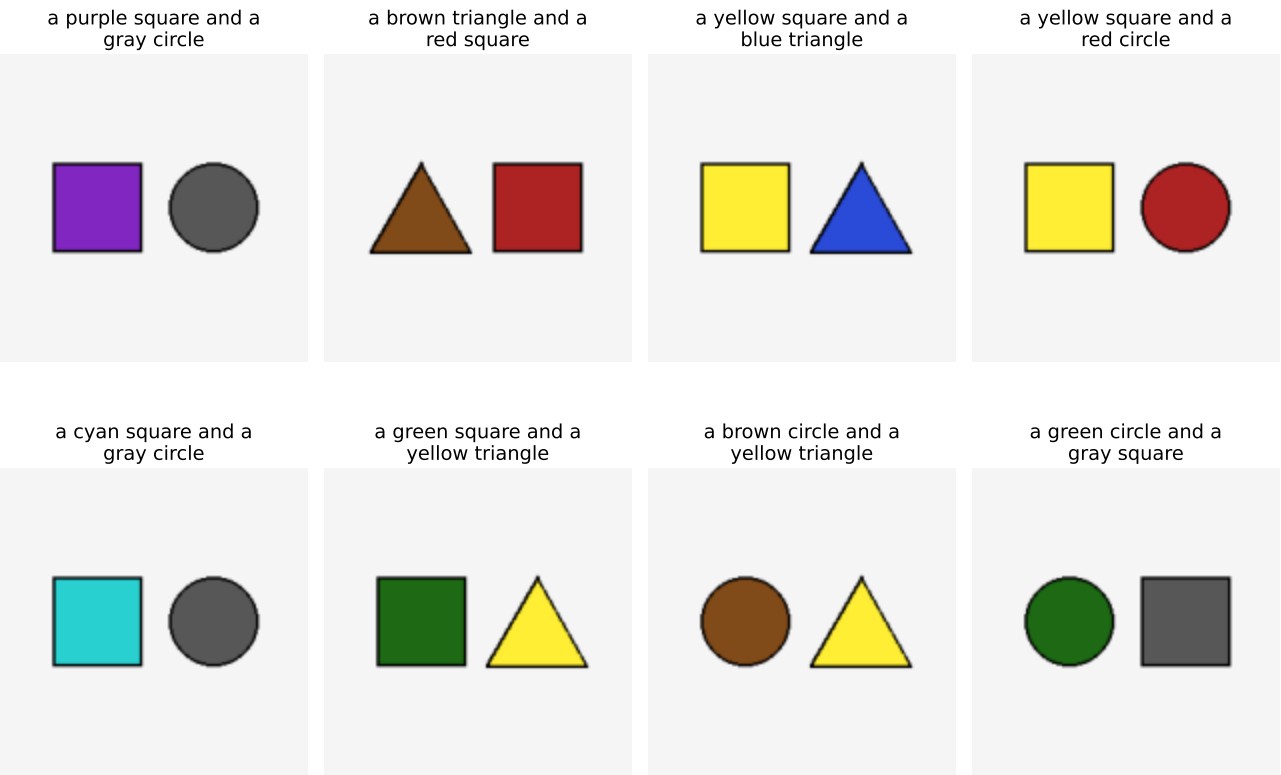

*Figure 12.* **CLEVR-2D dataset samples.** Our 2D adaptation of CLEVR, replacing 3D shapes with flat 2D equivalents. Same concept structure as CLEVR (8 colors × 3 shapes, 576 scenes).

## C.2. CLIP geometry experiments

This section specifies how we compute the *retrieval* and *probing* metrics reported for the Level-I/II decomposition experiments (Tab. 2) and for the intervention experiments (§4.1). To evaluate whether binding-relevant information is present in

embeddings *within a modality*, we use *trained linear probes* on frozen embeddings.

**Probing metric (concept/object recognition accuracy).** Let $z_s = f(x_s) \in \mathbb{R}^d$ denote an embedding of a scene $s$ (image or text, depending on the encoder). For each concept $i \in [C]$, we train a linear probe $h_i(z) = W_i z + b_i \in \mathbb{R}^{|C_i|}$ that outputs logits over the values of concept $i$. We also train an object probe $h_{\text{obj}}(z) = W_{\text{obj}} z + b_{\text{obj}} \in \mathbb{R}^{|C|}$ that outputs logits over the object space (concept tuples). Because scenes contain multiple objects, the target for a probe is generally *multi-hot* (multiple values/objects are present). We train probes on frozen embeddings by minimizing cross-entropy to the normalized multi-positive target distribution (as in §B).

At evaluation time, we evaluate using the metrics in Def. 3.3 and Def. 3.4. For concept $i$, let $h_i(z_s)_v$ be the logit for value $v \in C_i$. We count the scene as correct for concept $i$ if

$$\min_{v \in V_i(s)} h_i(z_s)_v > \max_{v \in C_i \setminus V_i(s)} h_i(z_s)_v.$$

Concept recognition accuracy is the mean of this indicator over scenes and then averaged across concepts. Object recognition accuracy is computed analogously from $h_{\text{obj}}(z_s)$, using the set of objects $O(s)$.

**Retrieval metric (nearest-neighbor scene retrieval).** For a fixed dataset of two-object scenes $\{s_n\}_{n=1}^N$, we precompute their embeddings $z_n = f(x_{s_n})$. Given a predicted embedding $\tilde{z}_n$ (e.g., a reconstruction from object means, or an edited embedding from §4.1), retrieval is treated as a nearest-neighbor problem in embedding space:

$$\hat{n} = \arg \max_{m \in [N]} \frac{\langle \tilde{z}_n, z_m \rangle}{\|\tilde{z}_n\|_2 \|z_m\|_2}.$$

Retrieval is correct if the nearest neighbor corresponds to the intended scene. When multiple scenes share the same underlying object tuple (e.g., because of positional variation), we treat any scene with the intended object tuple as correct. We report top-1 accuracy over all evaluated scenes.

**Reconstruction metric.** For the $R^2$ score, we measure how much variance is explained by the addition of the two object embeddings. In other words, the score measures how well $V^2$ vectors can reconstruct all of the $V^4$ embeddings. The $R^2$ score is defined as

$$R^2 := 1 - \frac{\sum_{x_s} \|f(x_s) - u_{o_0} - u_{o_1}\|_2^2}{\sum_{x_s} \|f(x_s) - f\|_2^2}. \tag{7}$$

### C.3. Text encoder decomposition

**Setup.** We generate a synthetic caption dataset for scenes by specifying a finite *concept space*

$$\mathcal{C} := \mathcal{C}_1 \times \mathcal{C}_2,$$

where $\mathcal{C}_1$ contains *adjective* tokens (e.g., color/size/material) and $\mathcal{C}_2$ contains *noun* tokens (e.g., shape/category). In our experiments we generate 10,000 two-object scenes plus 100 single-object scenes; each scene is a list of objects, and each object is represented by a pair of integer indices $(i_1, i_2)$ selecting tokens from $\mathcal{C}_1$ and $\mathcal{C}_2$.

**Caption construction.** For each scene, we linearize objects into a string of the form "a $\langle$attr$\rangle$ $\langle$obj$\rangle$ and a $\langle$attr$\rangle$ $\langle$obj$\rangle$". The following snippet shows the exact procedure used.

```
concept_space = [
    ["red", "blue", "green", "yellow", "purple", "orange", "pink", "brown", "gray", "black
        ",
     "small", "large", "tiny", "round", "sharp", "rough", "bumpy", "smooth", "matte", "
        glossy"],
    ["circle", "square", "triangle", "star", "heart", "spiral", "sphere", "cylinder", "
        cone", "pyramid",
     "car", "cat", "dog", "man", "woman", "tree", "flower", "rock", "bird", "fish"],
]

captions_dataset = []

for scene in scene_dataset:
    scene_caption = "a"
```

```
    for obj_i, obj in enumerate(scene):
        for concept_i, concept in enumerate(obj):
            scene_caption += f"␣{concept_space[concept_i][concept]}"
        if obj_i != len(scene) - 1:
            scene_caption += "␣and␣a"
    captions_dataset.append(scene_caption)
```

### C.4. Details on approximating pre-trained models' binding function

This section provides experimental details for the binding function approximation experiments on pre-trained models (§5.1). Our aim is to determine whether CLIP's binding function *can* be approximated by a learnable function, rather than to find the most efficient approximator. To ensure that any failure to generalize reflects genuine complexity of the binding function rather than insufficient model capacity, we deliberately over-parameterize the approximator and sweep over a wide range of configurations.

**MLP architectures.** We use ReLU MLPs that take as input a concatenation of one-hot encodings of concept indices for both objects in the scene (i.e., $2 \times C \times V$ binary features, where $C$ is the number of concepts and $V$ is the number of values per concept). We experiment with 1-layer MLPs with hidden widths $\in \{64, 128, 256, 512, 1024, 2048, 4096\}$;

The output dimension matches the embedding dimension $d$ of the encoder (e.g., $d = 512$ for CLIP ViT-B/32). Results reported in the main text correspond to the *best* performance achieved across all architectures for each training fraction, ensuring that we report an upper bound on what the approximator family can achieve.

**Training.** We use the Adam (Kingma & Ba, 2017) optimizer with learning rates in $\{10^{-2}, 10^{-3}, 10^{-4}\}$, batch size 4096, for 10,000 steps. We select the best configuration based on test object accuracy. We split the $V^C$ possible objects into train/test sets, varying training coverage over $\{10\%, 20\%, \ldots, 90\%\}$. Test scenes contain only held-out objects, i.e., objects whose concept configurations never appeared during training.

**Evaluation.** We apply linear probes (trained on true embeddings) to the predicted embeddings and measure concept/object recognition accuracy. This tests whether predicted embeddings preserve functional binding properties, not just reconstruction fidelity. For text: we use synthetic captions (§C.3); for images: CLEVR, CLEVR-2D, PUG:SPARE. We report the best accuracy across all MLP architectures and learning rates for each configuration.

## D. Additional results

### D.1. Interventions

In §4.2, we test how concept and object components each contribute to concept and object recognition in scene embeddings.

**Setup.** We estimate object and concept embeddings as described in Tab. 6. We use the same retrieval and probing metrics defined in §C.2. For interventions, the *target* of an edited embedding is the intended counterfactual scene (or, when multiple scenes share the same object tuple due to positional variation, any scene with that object tuple is treated as correct).

**Permuted-ID controls.** To ensure that effects are not explained by subtracting an arbitrary but similarly-sized signal, we also run *permuted-ID* controls for both concept and object ablations. Concretely, let $N_\mathrm{c} := \sum_{i=1}^{C} |\mathcal{C}_i|$ be the total number of concept values (often $N_\mathrm{c} = C \cdot V$), and let $N_\mathrm{o} := |\mathcal{C}| = \prod_{i=1}^{C} |\mathcal{C}_i|$ be the number of objects (concept tuples). We sample uniform random permutations $\pi_\mathrm{c} : [N_\mathrm{c}] \to [N_\mathrm{c}]$ and $\pi_\mathrm{o} : [N_\mathrm{o}] \to [N_\mathrm{o}]$. When applying $\pi_\mathrm{c}$, we treat each concept value $(i, v)$ as a unique ID in $[N_\mathrm{c}]$; when applying $\pi_\mathrm{o}$, we treat each object $o \in \mathcal{C}$ as a unique ID in $[N_\mathrm{o}]$. We then form the same residual embeddings as in the main text, but with permuted components, e.g.,

$$f(\boldsymbol{x_s}) - \sum_i \boldsymbol{u}_{\pi_\mathrm{c}(c_{1,i})} - \sum_i \boldsymbol{u}_{\pi_\mathrm{c}(c_{2,i})} \quad \text{or} \quad f(\boldsymbol{x_s}) - \boldsymbol{u}_{\pi_\mathrm{o}(\boldsymbol{o}_1)} - \boldsymbol{u}_{\pi_\mathrm{o}(\boldsymbol{o}_2)}.$$

We evaluate these permuted residuals with the same probes.

*Table 10.* **Decodability results.** We report probe accuracy after subtracting components from the embedding (remove concepts/objects) or their permuted-ID controls, for the image encoder and its random-init counterpart for nonpositional / positional variations.

| Intervention | Conc. (↑) | Obj. (↑) |
|---|---|---|
| **CLIP ViT-B/32 (CLEVR-2D)** | | |
| None | *1.00 / 1.00* | *0.99 / 0.99* |
| Subtract concepts | *0.32 / 0.30* | *0.78 / 0.79* |
| Subtract objects | *0.28 / 0.29* | *0.08 / 0.10* |
| Permute concepts | *0.83 / 0.83* | *0.95 / 0.95* |
| Permute objects | *0.92 / 0.91* | *0.86 / 0.86* |
| **CLIP ViT-B/32 (CLEVR)** | | |
| None | *1.00 / 1.00* | *0.93 / 0.93* |
| Subtract concepts | *0.28 / 0.27* | *0.77 / 0.79* |
| Subtract objects | *0.26 / 0.23* | *0.01 / 0.03* |
| Permute concepts | *0.83 / 0.81* | *0.91 / 0.93* |
| Permute objects | *0.93 / 0.91* | *0.84 / 0.82* |
| **DINOv2 ViT-B/14 (CLEVR-2D)** | | |
| None | *1.00 / 1.00* | *0.97 / 0.96* |
| Subtract concepts | *0.26 / 0.27* | *0.79 / 0.79* |
| Subtract objects | *0.25 / 0.23* | *0.03 / 0.06* |
| Permute concepts | *0.79 / 0.78* | *0.94 / 0.93* |
| Permute objects | *0.91 / 0.90* | *0.85 / 0.86* |
| **DINOv2 ViT-B/14 (CLEVR)** | | |
| None | *0.99 / 0.99* | *0.92 / 0.92* |
| Subtract concepts | *0.20 / 0.20* | *0.61 / 0.65* |
| Subtract objects | *0.18 / 0.20* | *0.01 / 0.00* |
| Permute concepts | *0.77 / 0.76* | *0.92 / 0.92* |
| Permute objects | *0.87 / 0.88* | *0.83 / 0.81* |

## D.2. Editing scene embeddings with object embeddings

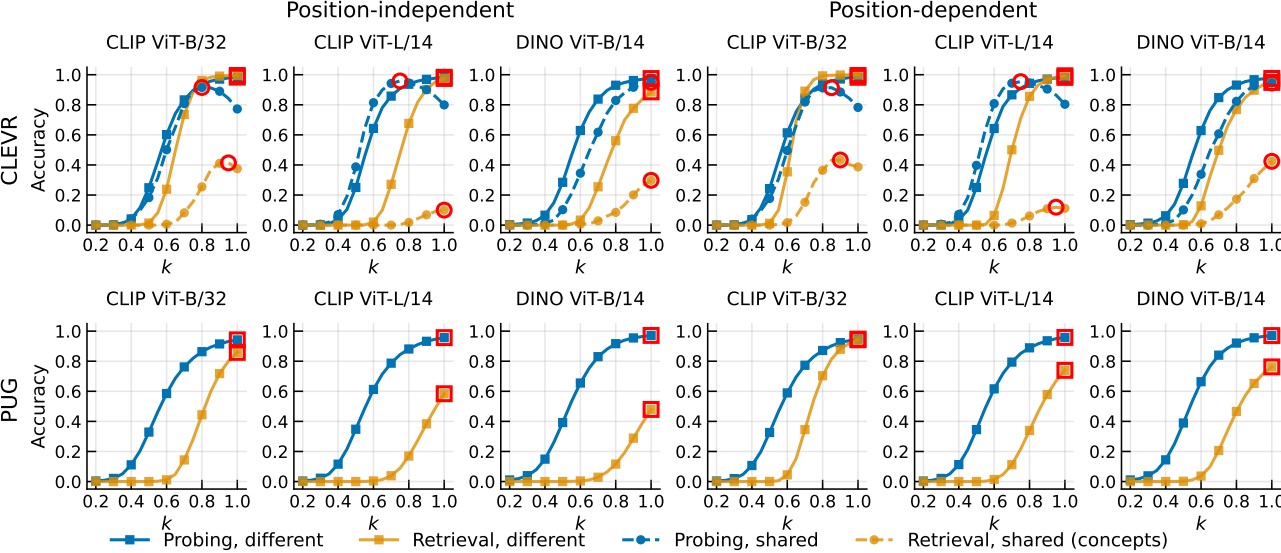

*Figure 13.* **Effect of intervention strength on object replacement.** We vary intervention strength $k$ with position-independent (AVG) and position-dependent (AVG+POS) object embeddings for CLEVR and PUG:SPARE. We distinguish between samples in which two objects have different concepts (solid line) and those with a shared concept (dashed line).

*Table 11.* **Decodability results.** We report probe accuracy after subtracting components from the embedding (remove concepts/objects) or their permuted-ID controls, for both text and image encoders and their random-init counterparts for nonpositional / positional variations on the PUG:SPARE dataset.

| Model | Intervention | Text | | Image | |
| | | Conc. (↑) | Obj. (↑) | Conc. (↑) | Obj. (↑) |
|---|---|---|---|---|---|
| Random | None | 0.95 / 0.95 | 0.59 / 0.60 | 0.78 / 0.78 | 0.71 / 0.71 |
| | Subtract concepts | 0.02 / 0.02 | 0.27 / 0.43 | 0.06 / 0.08 | 0.66 / 0.68 |
| | Subtract objects | 0.01 / 0.01 | 0.00 / 0.00 | 0.04 / 0.05 | 0.00 / 0.01 |
| | Permute concepts | 0.88 / 0.88 | 0.69 / 0.75 | 0.80 / 0.81 | 0.78 / 0.76 |
| | Permute objects | 0.83 / 0.84 | 0.93 / 0.98 | 0.78 / 0.80 | 0.88 / 0.88 |
| CLIP-B/32 | None | 1.00 / 1.00 | 1.00 / 1.00 | 0.95 / 0.95 | 0.88 / 0.88 |
| | Subtract concepts | 0.07 / 0.07 | 0.99 / 0.99 | 0.06 / 0.08 | 0.73 / 0.73 |
| | Subtract objects | 0.05 / 0.05 | 0.02 / 0.03 | 0.04 / 0.04 | 0.00 / 0.00 |
| | Permute concepts | 0.92 / 0.93 | 1.00 / 1.00 | 0.84 / 0.84 | 0.91 / 0.91 |
| | Permute objects | 0.96 / 0.96 | 1.00 / 1.00 | 0.82 / 0.83 | 0.92 / 0.93 |
| DINOv2-B/14 | None | - | - | 0.95 / 0.94 | 0.79 / 0.79 |
| | Subtract concepts | - | - | 0.04 / 0.04 | 0.62 / 0.70 |
| | Subtract objects | - | - | 0.03 / 0.04 | 0.00 / 0.00 |
| | Permute concepts | - | - | 0.77 / 0.77 | 0.80 / 0.79 |
| | Permute objects | - | - | 0.83 / 0.83 | 0.87 / 0.87 |

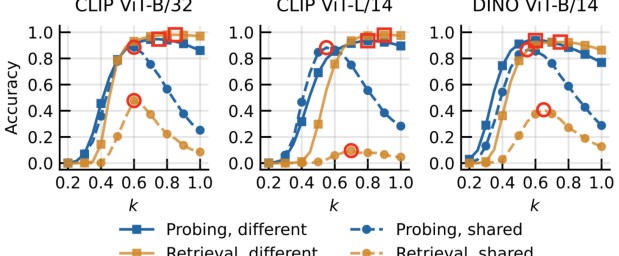

*Figure 14.* **Object editing with SINGLE-OBJ embeddings.** Intervention strength $k$ is varied for CLEVR image embeddings using object embeddings estimated from single-object scenes.

*Figure 15.* **Object editing for text embeddings.** We vary the intervention strength $k$ of object replacement for text encoder embeddings.

We provide additional results for the object-replacement interventions introduced in §4.1. These experiments test whether object embeddings can be reused across scenes to produce counterfactual scene embeddings via simple arithmetic.

**Setup.** Given a two-object scene $s = (o_1, o_2)$ with embedding $f(x_s)$, we construct an edited embedding by replacing $o_1$ with a counterfactual object $o'_1$:

$$\tilde{z} = f(x_s) - k \, u_{o_1} + k \, u_{o'_1}, \tag{8}$$

where $k$ controls the strength of the intervention. Setting $k = 0$ leaves the embedding unchanged, while larger $k$ increases the contribution of the replacement. In practice, performance varies with $k$, and we report full curves over $k$ in Figure 13. Unless stated otherwise, results in Tab. 12 use $k = 1.0$.

We distinguish between two settings. In the *different-concepts* setting, the two objects in the scene do not share any concept values (e.g., a red cube and a blue sphere). In the *shared-concepts* setting, the objects share at least one concept value (e.g., a red cube and a red sphere). The shared-concepts setting is more challenging because the two objects occupy overlapping concept directions in the representation.

**Results.** Table 12 reports probing and retrieval performance for object replacement interventions at full strength ($k = 1.0$) across datasets and models. In most settings, replacing an object embedding yields scene embeddings that support both object probing and retrieval consistent with the intended counterfactual scene. Object embeddings obtained via scene averaging and position-specific averaging perform similarly across datasets, indicating that positional effects are limited. When available, embeddings derived from single-object scenes also enable effective edits, though with reduced performance in some settings.

*Table 12.* **Intervention results summary at** $k = 1.0$**.** Object embeddings are constructed by object averaging (AVG), position-wise averaging (AVG+POS), or from single-object scenes (SINGLE-OBJ).

| Dataset | Model | Concepts | Probing (↑) | | | Retrieval (↑) | | |
|---|---|---|---|---|---|---|---|---|
| | | | AVG | AVG+POS | SINGLE-OBJ | AVG | AVG+POS | SINGLE-OBJ |
| Text | CLIP ViT-B/32 | Different | 0.99 | 0.99 | 0.69 | 0.99 | 0.99 | 0.96 |
| Text | CLIP ViT-B/32 | Shared | 0.86 | 0.86 | 0.23 | 0.73 | 0.73 | 0.10 |
| CLEVR | CLIP ViT-B/32 | Different | 0.98 | 0.98 | 0.86 | 1.00 | 1.00 | 0.97 |
| CLEVR | CLIP ViT-B/32 | Shared | 0.77 | 0.78 | 0.25 | 0.38 | 0.39 | 0.08 |
| CLEVR | CLIP ViT-L/14 | Different | 0.98 | 0.98 | 0.90 | 0.98 | 0.99 | 0.98 |
| CLEVR | CLIP ViT-L/14 | Shared | 0.80 | 0.80 | 0.28 | 0.10 | 0.11 | 0.05 |
| CLEVR | DINO ViT-B/14 | Different | 0.97 | 0.98 | 0.77 | 0.89 | 0.95 | 0.86 |
| CLEVR | DINO ViT-B/14 | Shared | 0.95 | 0.94 | 0.29 | 0.30 | 0.42 | 0.13 |
| CLEVR-2D | CLIP ViT-B/32 | Different | 0.98 | 0.98 | 0.92 | 0.99 | 0.99 | 0.97 |
| CLEVR-2D | CLIP ViT-B/32 | Shared | 0.68 | 0.67 | 0.26 | 0.14 | 0.14 | 0.05 |
| CLEVR-2D | CLIP ViT-L/14 | Different | 0.98 | 0.98 | 0.90 | 0.95 | 0.95 | 0.90 |
| CLEVR-2D | CLIP ViT-L/14 | Shared | 0.72 | 0.72 | 0.36 | 0.13 | 0.14 | 0.07 |
| CLEVR-2D | DINO ViT-B/14 | Different | 0.97 | 0.98 | 0.72 | 0.97 | 0.97 | 0.88 |
| CLEVR-2D | DINO ViT-B/14 | Shared | 0.87 | 0.88 | 0.29 | 0.14 | 0.16 | 0.03 |
| PUG:SPARE | CLIP ViT-B/32 | Different | 0.94 | 0.95 | – | 0.86 | 0.94 | – |
| PUG:SPARE | CLIP ViT-L/14 | Different | 0.96 | 0.96 | – | 0.58 | 0.74 | – |
| PUG:SPARE | DINO ViT-B/14 | Different | 0.97 | 0.97 | – | 0.48 | 0.76 | – |

Performance drops substantially when the two objects in the scene share concepts, particularly for retrieval. This effect is consistent across datasets and encoders. A plausible explanation is feature interference: when objects share concept values, their contributions overlap in representation space, making it harder to selectively remove and insert a single object without affecting the other (Campbell et al., 2024).

We further analyze the role of intervention strength in Figs. 13 to 15. Varying the intervention coefficient $k$ reveals that performance typically improves with stronger interventions before saturating. However, in shared-concept settings, optimal performance is often achieved at smaller values of $k$, suggesting that partial edits mitigate interference effects.

### D.3. Three-object scenes and occlusions

To check that the additive decomposition of §4.1 is not limited to two-object scenes, we generate CLEVR scenes with three objects and matching text descriptions (Fig. 16). We evaluate $R^2$ reconstruction, probing accuracy, and retrieval as in Tab. 2 and Tab. 3.

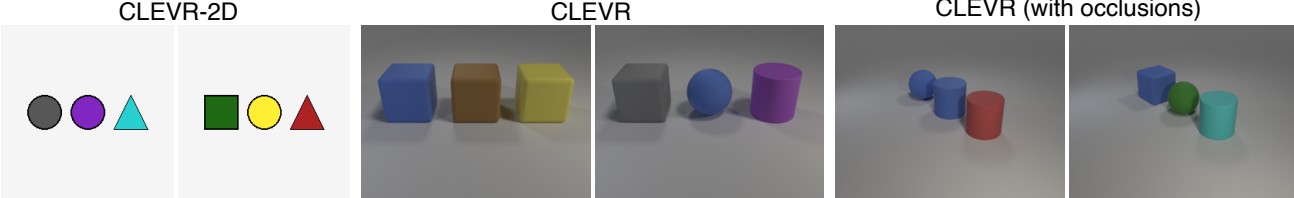

*Figure 16.* **Example CLEVR scenes with 3 objects.** Used for the 3-object decomposition evaluation in Tab. 13.

*Table 13.* **Object decomposition extends to 3-object scenes (Fig. 16).** $R^2$ reconstruction from concepts and objects, probing accuracy, and retrieval accuracy for scenes with 3 objects (as in Tab. 2 and Tab. 3). Reconstruction from objects remains stronger than from concepts. Occlusions degrade retrieval but probing stays high.

| Dataset | Concepts $R^2$ | Objects $R^2$ | Probing | Retrieval |
|---|---|---|---|---|
| Text | 0.83 | 0.86 | 0.86 | 0.93 |
| CLEVR | 0.67 | 0.72 | 0.93 | 0.76 |
| CLEVR (occlusions) | 0.65 | 0.70 | 0.91 | 0.65 |
| CLEVR-2D | 0.67 | 0.73 | 0.82 | 0.76 |

The additive decomposition extends to 3 objects: probing accuracy remains 0.91–0.93 on CLEVR even with occlusions, and reconstruction from objects consistently exceeds reconstruction from concepts, matching the pattern in Tab. 2. Retrieval drops under occlusion (0.76 → 0.65), as expected, consistent with prior findings that VLM binding degrades as the number of objects grows (Campbell et al., 2024).

### D.4. Extension to natural images

**Setup.** To assess whether the additive decomposition extends beyond synthetic datasets, we generate images using Gemini Nano Banana 2 (gemini-3.1-flash-image-preview), spanning 5 object types and 5 patterns (625 samples). Unlike the synthetic datasets, these images exhibit variation in size, shape, color, and pattern realization, making the task substantially more challenging. We evaluate object decomposition and editing following the same protocol as in Tab. 2 and Tab. 3, using retrieval-based metrics: full retrieval among all 625 scenes, 4-way retrieval (correct scene, permuted concept combination, and two random distractors), and binary retrieval between the original and edited scene embeddings.

| striped mug and polka dot umbrella | floral vase and floral vase | zigzag umbrella and polka dot chair | striped backpack and floral chair | checkerboard backpack and zigzag mug | zigzag chair and zigzag vase |

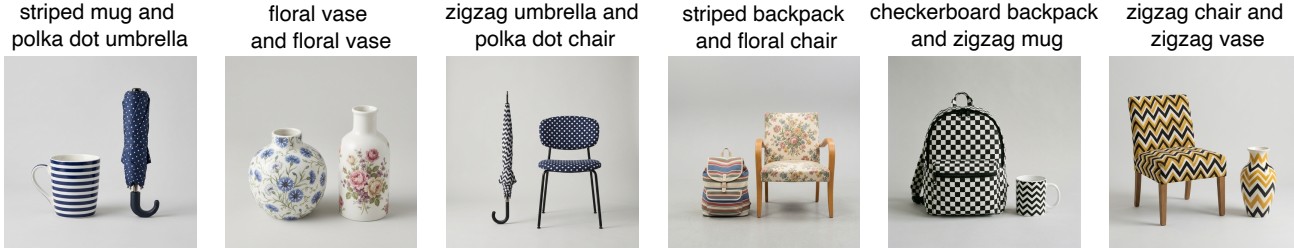

*Figure 17.* **Example images spanning 5 objects and 5 patterns.** Images were generated with Gemini Nano Banana 2 (gemini-3.1-flash-image-preview) at $512 \times 512$ resolution using the prompt template: *"A square studio product photo of exactly two objects: a {obj1} with a {pattern1} print on its surface, next to a {obj2} with a {pattern2} print on its surface. Exactly two objects only. The pattern must appear only as the surface design of each object, not as separate objects, contents, decorations, or background elements."* Objects: mug, backpack, umbrella, vase, chair. Patterns: striped, polka dot, checkerboard, floral, zigzag. This yields $5^2 \times 5^2 = 625$ unique two-object scenes. The images more closely resemble natural images; objects vary in size, shape, and color, and patterns are realized differently across samples. See Tab. 14 for results.

**Results.** Results are well above chance despite the increased difficulty, suggesting that the additive structure extends, to some degree, to more naturalistic scenes. We note that with a larger sample size, object embedding estimates would likely improve further.

*Table 14.* **Object decomposition and editing on the natural images.** We evaluate additive decomposition (as in Tab. 2) and object editing (as in Tab. 3) via retrieval on the 625 generated scenes. Results are above chance despite the increased visual complexity.

| Task | Setting | Retrieval | Random chance |
|------|---------|-----------|---------------|
| Decomposition | All (625) | 54% | $1/625 \approx 0.16\%$ |
| Decomposition | 4-way | 81% | 25% |
| Object editing | All (625) | 68% | $1/625 \approx 0.16\%$ |
| Object editing | Binary | 95% | 50% |

### D.5. MDS visualization

To build intuition about the embedding geometry, we visualize MDS projections for a small set of concepts (Borg & Groenen, 2005). We consider two shapes (C, S) and two colors (R, B). This yields objects such as RC ('red cube') and BS ('blue sphere'), and two-object scenes such as RCBS ('red cube and blue sphere'). Object and concept embeddings are estimated as averages of embeddings of scenes that contain the corresponding component. Purple points represent sums of object embeddings (e.g. RC+BS). We connect the embeddings of the corresponding two-object scenes (e.g. RCBS) to these summed vectors.

**Results.** Across datasets and models, the projections exhibit a partial grid-like structure reflecting color and shape composition. This structure is most pronounced for text embeddings, while for CLIP image embeddings, it is more distorted. In DINOv2, objects with different shapes are far apart in embedding space, whereas objects with similar colors remain close.

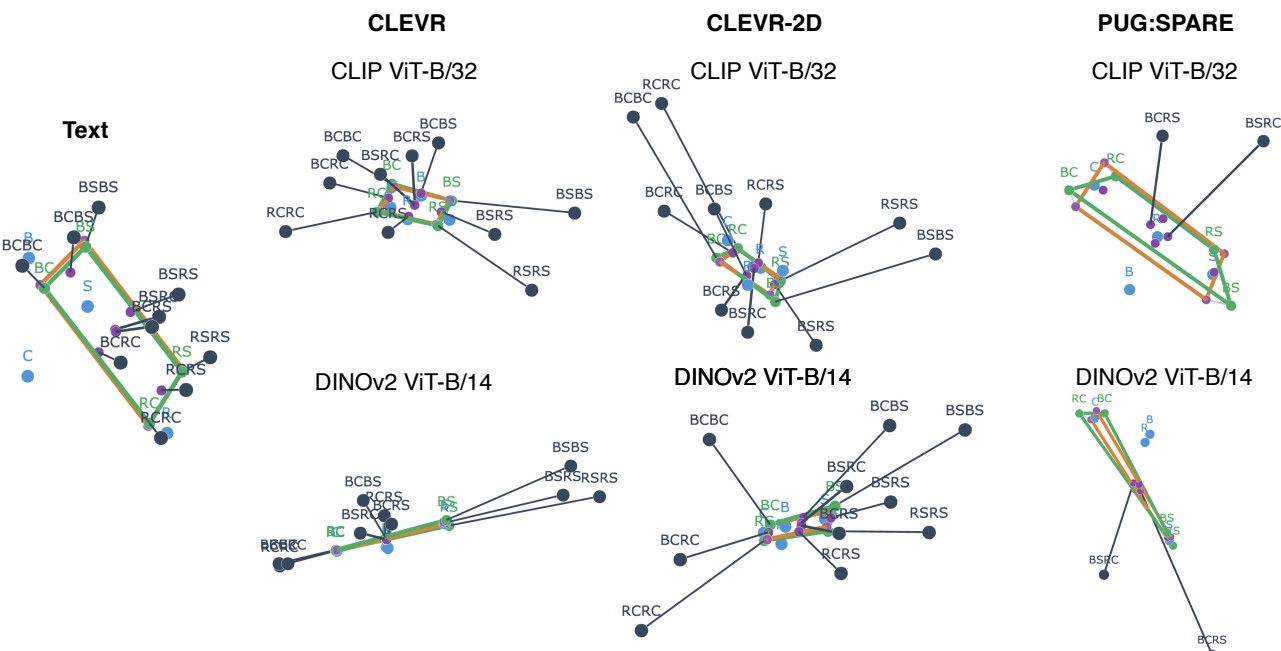

*Figure 18.* **MDS shows an approximate additive structure using averaged object embeddings.** MDS projections of CLIP and DINOv2 embeddings for scenes from CLEVR, CLEVR-2D, and PUG (distances approximate embedding distances). Object embeddings (e.g. RC for 'red cube') and concept embeddings (e.g. R for 'red') are estimated as averages of embeddings of scenes containing the corresponding component. Two-object scenes are denoted by concatenation (e.g. RCBS for 'red cube and blue sphere'). Purple points correspond to sums of object embeddings (e.g. RC+BS). Line segments connect these summed vectors to the corresponding two-object scene embeddings.

## D.6. Robustness of the binding-complexity diagnostic to the approximator family

The binding-complexity analyses in §5.1 and §5.3 uses a 1-hidden-layer MLP as the approximator. Our choice is motivated by the view that MLPs trained with SGD tend to favor simple, compressible solutions (Blier & Ollivier, 2018; Wilson, 2025), and that compositional mappings are precisely the kind of solutions this bias should favor (Ren & Sutherland, 2024). If a simple compositional binding function exists, this approximator family should find it.

To verify that our findings are not specific to the MLP family, we additionally fit XGBoost and Random Forest approximators with the same setup as §5.1 (predict CLIP/DINOv2 scene embeddings from discrete concept indices, evaluate on held-out objects). For XGBoost we sweep `n_estimators` $\in \{100, 500, 1000\}$, `max_depth` $\in \{6, 8, 16, 32\}$, `lr` $\in \{0.1, 0.01\}$. For Random Forest we sweep `n_estimators` $\in \{100, 500, 1000\}$, `max_depth` $\in \{\text{None}, 8, 16, 32\}$. We report the best accuracy across configurations.

*Table 15.* **Approximator family does not change the conclusion.** Best object / concept recognition accuracy on held-out objects when approximating scene embeddings from concept indices, for different approximator families. Across all three families, concept recognition is high while object recognition stays near zero, indicating that the bottleneck is the binding structure rather than the choice of approximator.

| Approximator | CLIP B/32 (img) Obj. / Conc. | DINOv2 B/16 (img) Obj. / Conc. | CLIP B/32 (txt) Obj. / Conc. |
|---|---|---|---|
| MLP | 0.17 / 0.97 | 0.10 / 1.00 | 0.21 / 1.00 |
| XGBoost | 0.00 / 0.94 | 0.03 / 1.00 | 0.18 / 1.00 |
| RF | 0.00 / 0.75 | 0.00 / 1.00 | 0.04 / 0.90 |

In all three cases the approximator succeeds at concept-level prediction but fails at object-level binding. MLPs perform slightly better than XGBoost and Random Forest, though this is not an exhaustive search. The qualitative conclusion of §5.1 is unchanged: CLIP's binding function does not generalize to unseen objects across this set of approximator families.

### D.7. Multiplicative probe applied to CLIP and DINOv2

We complement §5.4 by fitting the same multiplicative Global product probe to CLIP and DINOv2 embeddings (Fig. 19). Concept recognition recovers with more training data, but object recognition stays near zero across encoders. Unlike the from-scratch models (§5.4, §E), neither CLIP nor DINOv2 admits a multiplicative binding function, consistent with their known cross-modal binding failures.

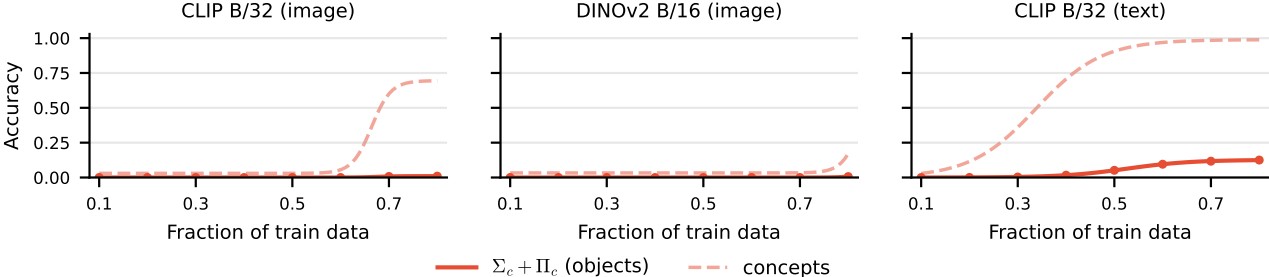

*Figure 19.* **Multiplicative structure does not hold for CLIP and DINOv2.** We fit the Global product probe on CLIP and DINOv2 embeddings. Concept recognition (dashed) recovers with more training data, but object recognition (solid) stays near zero for all encoders. Unlike the from-scratch models trained on pixels (Fig. 21, bottom), CLIP's and DINOv2's binding functions are too complex to be captured by the multiplicative probe, consistent with their failure to bind concepts cross-modally.

## E. Vision encoders trained from scratch

The controlled experiments in §5.2–§5.4 use multi-hot token sequences as input to the scene encoder. To verify that the conclusions are not an artifact of the discrete input format, we replicate the entire pipeline (§5.2–§5.4) with *pixel* inputs and a from-scratch vision encoder.

**Setup.** We replace the multi-hot embedding layer of the scene encoder with a convolutional front-end and increase the number of transformer layers from 6 to 8 (we found this configuration to generalize better). The training pipeline, evaluation protocol, and binding analyses are otherwise identical to the main paper. Each object is defined by two concepts (square color and border color) with $V = 50$ values, giving up to $6.5 \times 10^6$ object combinations. To stress the visual difficulty, we consider three levels of pixel-space complexity (Fig. 20): (i) noise-free, non-overlapping objects; (ii) speckled noise with non-overlapping objects; and (iii) noisy and overlapping objects.

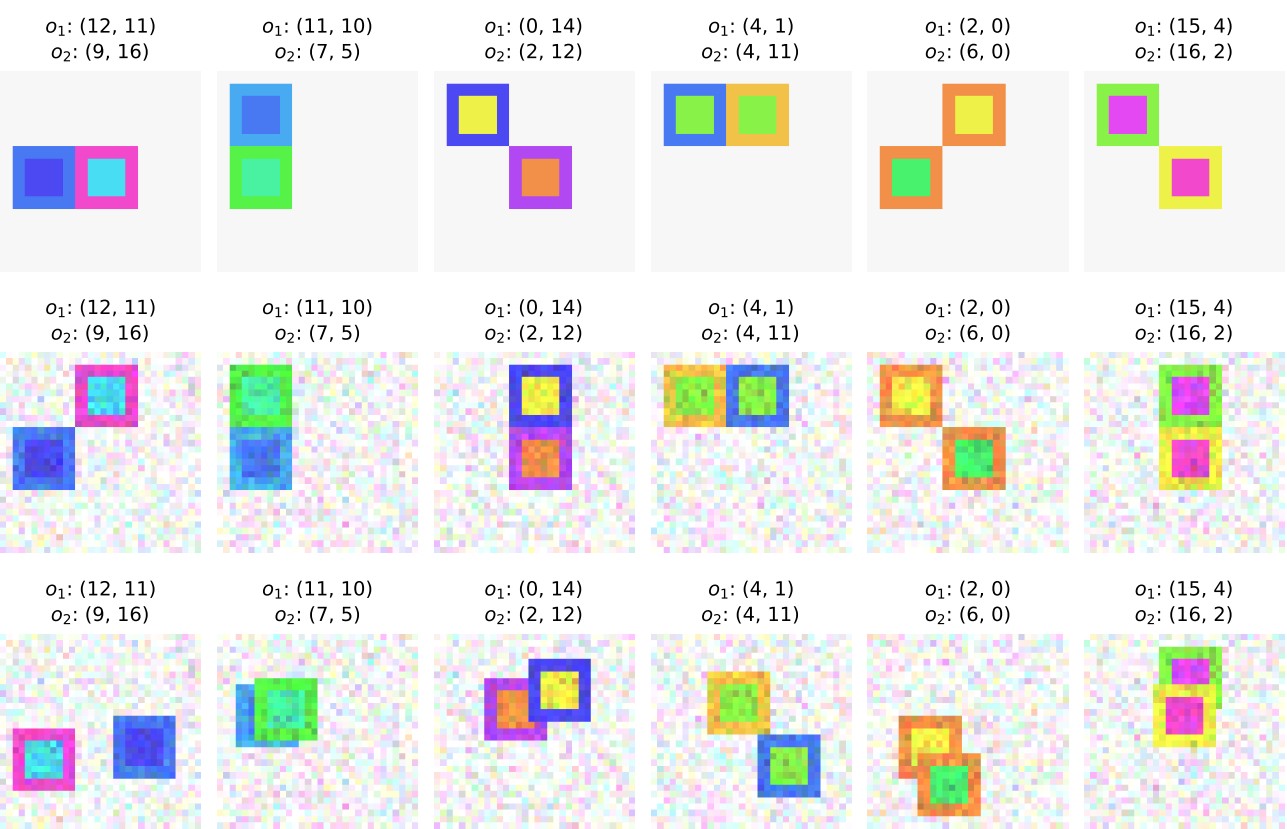

*Figure 20.* **Synthetic pixel samples used to train vision encoders from scratch.** Three levels of pixel-space complexity: (top) noise-free, non-overlapping objects; (middle) speckled noise, non-overlapping objects; (bottom) noisy and overlapping objects. Each object is defined by two concepts (square color and border color). Samples shown correspond to the $C = 2$, $V = 50$ setting, giving up to $6.5 \times 10^6$ object combinations; the number of unique scenes is larger due to random positions and noise. See Fig. 21 for the corresponding binding-complexity and multiplicative-probe results.

**Results.** The two main findings from §5.3–§5.4 replicate on pixel inputs (Fig. 21). The binding maps of generalizing from-scratch vision models are well approximated by shallow MLPs (top row), and the multiplicative Global product probe recovers object recognition on held-out objects (bottom row), even under speckled noise and overlapping objects. Contrast with Fig. 19, where the same probe fails on the pretrained CLIP and DINOv2 encoders.

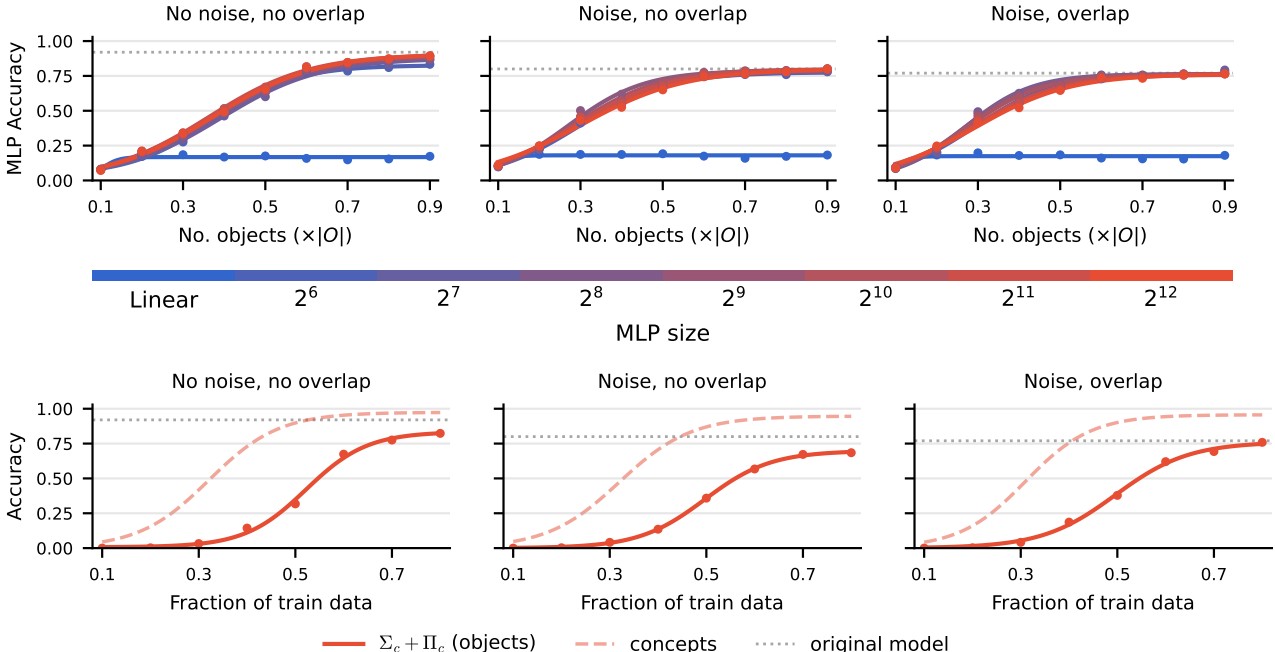

*Figure 21.* **Vision-domain results corresponding to Fig. 20.** Columns correspond to the three visual complexity levels. **(Top)** Binding-complexity results, analogous to §5.3: even small MLPs approximate the binding function well, confirming that from-scratch vision encoders learn low-complexity binding functions. **(Bottom)** Multiplicative-probe results, analogous to §5.4: the Global product probe recovers object recognition on held-out objects, confirming that the multiplicative structure observed on multi-hot inputs extends to the vision setting. Contrast with Fig. 19, where the same probe fails on the pretrained CLIP and DINOv2 encoders.

