# OpenReview forum: "How can embedding models bind concepts?"
_ICML.cc/2026/Conference — ICML 2026 spotlight_

### Official Review · Reviewer_7ZcV · 2026-03-10

**Soundness:** 2
**Presentation:** 2
**Significance:** 2
**Originality:** 2
**Overall Recommendation:** 4
**Confidence:** 4

**Summary:**

This paper studies concept binding in embedding models, focusing mainly on CLIP. The authors distinguish between concept recognition and object recognition, and argue that although CLIP embeddings contain object-related information and show an additive object-level structure, the mapping from concepts to object/scene embeddings is too complex to generalize well to unseen concept combinations. To support this, they analyze CLIP and DINOv2 on synthetic multi-object datasets, and then train controlled CLIP-style transformer encoders from scratch on toy synthetic data, showing that in this controlled setting, binding can generalize and is better explained by low-complexity, multiplicative interactions between concepts.

**Compliance With Llm Reviewing Policy:**

Affirmed.

**Final Justification:**

The rebuttal discussion was very helpful, and my main concerns were largely addressed. The authors clarified the intended scope of the paper, refined the interpretation of the binding-complexity results, and provided additional evidence that strengthened the main claims. I still think the final version should more clearly explain why the zero-shot setting studied here is of particular interest to the community, especially since in many practical applications these models are used together with downstream models rather than on their own. Overall, however, the rebuttal improved my assessment of the work, and I therefore recommend the acceptance of the paper and raise my score to 4.

**Key Questions For Authors:**

My questions are mainly around the weaknesses mentioned above:
1. Main contribution / take-away: What is the main novel insight that the authors would like the community to take from this paper? I would appreciate it if the authors could elaborate on what the implications of this work are.
2. On Section 5.1 vs 5.2: My understanding is that in Section 5.1 the input to the 1-layer MLP model is the discrete concept specification of the scene, and the target is the pretrained embedding of that scene, and the training objective is the MSE between the two models' outputs. In other words, one can interpret this as a distillation of CLIP representations into a 1-layer MLP and I am not sure if this is the correct setup for the paper's claims. This setup might make sense in the synthetic setting, since the concept tuple almost fully specifies the scene. However, I would appreciate it if the authors could better motivate why this is the right diagnostic for CLIP, and why success/failure of this approximator should be interpreted as explaining CLIP’s cross-modal binding behavior.
3. On the meaning of “high-complexity”: In Section 5.1, the paper concludes that CLIP’s binding function is high-complexity because a 1-layer MLP trained from concept indices does not generalize well to held-out objects. Could the authors clarify that this is a notion of complexity relative to the chosen approximator family? More broadly, why should failure of a 1-hidden-layer MLP be interpreted as evidence for high binding complexity, rather than simply a limitation of that approximator class?
4. Connection to real models: Sections 5.2–5.4 show that controlled transformer encoders trained on toy synthetic data can generalize binding and are well explained by multiplicative structure. Can the authors elaborate on how strongly they believe these results transfer to real pretrained models such as CLIP which are trained on a larger scale and on real-world data? At the moment, the controlled setup feels quite far from the natural image / natural language regime.
5. Related work: The paper seems to miss some closely related works on object-centric representations and compositional generalization in multi-object scenes. In particular, [1] studies compositional generalization of object-centric representations and pretrained SSL models such as DINOv2 and SigLip2, and [2] analyzes object-centric and pretrained foundation models on object property/concept prediction and VQA tasks. Could the authors discuss how these works relate to the present paper and how the current perspective differs from them?
6. Synthetic limitation / scale: The experiments focus on toy scenes with at most two objects. Do the authors expect the same conclusions to hold for more than two objects, where interference and combinatorial structure become more complex? How about when the objects are not perfectly recognizable and are entangled (e.g. presence of occlusion)? I would appreciate it if the authors could elaborate.
7. Section 5.4 / multiplicative structure: The multiplicative probe results are interesting, but they are again demonstrated only on the controlled models. Do the authors have any evidence that similar multiplicative structure helps explain the embeddings of pretrained models such as CLIP, or is this intended purely as a constructive existence proof?

[1] Kapl, Ferdinand, et al. "Are Object-Centric Representations Better At Compositional Generalization?." _arXiv preprint arXiv:2602.16689_ (2026).

[2] Mamaghan, Amir Mohammad Karimi, et al. "Exploring the Effectiveness of Object-Centric Representations in Visual Question Answering: Comparative Insights with Foundation Models." The Thirteenth International Conference on Learning Representations (2025).

**Limitations:**

See the questions and weaknesses above.

**Strengths And Weaknesses:**

### Strengths

1. The paper studies an interesting and important question: why vision-language embedding models can often recover object information within a modality, yet still fail at cross-modal binding on unseen concept combinations.

### Weaknesses

1. The paper’s main novelty, motivation, and practical implications are not fully clear. The work contains several ideas: additive decomposition of scene embeddings, a shallow approximator-based notion of “binding complexity,” and controlled toy transformer experiments showing emergence of generalization. Each part is somewhat interesting on its own, but the paper does not clearly explain what the main takeaway message is for the broader multimodal / representation learning community, or what practical implication follows beyond this specific binding setup.
2. The empirical setting is quite limited and is done on toy datasets, which weakens the connection back to real pretrained models such as CLIP. Most of the analysis is conducted on synthetic scenes/captions with at most two objects, fully known factorized concepts, and matching train/test generation procedures. In such a controlled setting, it is perhaps not too surprising that small transformers trained from scratch can learn systematic binding, or that simple multiplicative structures work well. While these experiments are still useful as proof-of-concept, it remains unclear how strongly the conclusions transfer to more realistic multi-object settings with natural images and text, where concepts are more entangled, less easily distinguishable, and not given in a clean factorized form.
3. The notion of “binding complexity” is operationalized in a narrow way. In Section 5.1, the claim that CLIP’s binding function is “high-complexity” is based on whether a 1-layer MLP can predict scene embeddings from concept indices and generalize to held-out objects. This is not a general measure of intrinsic complexity. In my humble opinion, a failure of this approximator family does not by itself imply that the underlying binding function is inherently high-complexity and it only shows that it is not easily captured by this simple approximator.
4. Some parts of the presentation feel under-polished or slightly disjoint. For example, the introduction is relatively short given the scope of the paper; the transition between the CLIP analysis and the controlled transformer study could be motivated more clearly; and some sections are harder to follow than necessary e.g. in Section 4.1, the paragraph titled “Effect of component removal on object recognition” appears before the actual results table is discussed in detail, and Table 3 is not referenced anywhere. Overall, the paper would benefit from a few rounds of polishing.

---

> ### Author Rebuttal · Authors · 2026-03-30
>
> Thank you for the thoughtful feedback. See the [rebuttal pdf](https://anonymous.4open.science/api/repo/anon12312321321313-B64A/file/rebuttal.pdf?v=f365b) for the results.
>
> **1. What is the main novel insight and implications?**
>
> The main takeaway is that cross-modal binding fails because the modalities produce high-complexity embeddings, while within each modality the embeddings exhibit object-based structure that enables uni-modal binding. Models that generalize can exist and typically exhibit low-complexity binding functions (e.g., multiplicative structure). The takeaway thus is that controlling the complexity of the binding mechanism is an important consideration for improving and understanding multimodal embedding models.
>
> **2. Why use discrete concept specifications as input to the MLP for CLIP? Why does its performance explain CLIP’s cross-modal binding?**
>
> - In standard evaluations, CLIP is tested on image-prompt pairs where the prompt (e.g., "red cat and blue dog") partially specifies the scene. Images contain details not mentioned in the text.
> - For the modalities to align, their embeddings must have high cosine similarity despite the information asymmetry.
> - The binding-relevant part of the embedding can be sufficiently specified with the discrete concept IDs.
> - Our MLPs tests precisely this: whether the binding can be approximated from concept specifications.
> - For the relation to cross-modal binding, please see EmAT, Q3.
>
> **3. Is "high-complexity" defined relative to the approximator family? Why does MLP failure indicate high binding complexity rather than limitations of that approximator?**
>
> Our notion of complexity is indeed relative to the approximator family. This is unavoidable, since computing (Kolmogorov) complexity is intractable. Approximating complexity via NN generalization is a standard approach: [1] uses 2-layer DNNs for the same purpose; we use 1-layer MLPs because we found 2 layers gave no difference while keeping the methodology cleaner.
>
> [1] Elmoznino et al., “A complexity-based theory of compositionality.” arXiv:2410.14817 (2024).
>
> **4. How well do the results from the controlled transformer setup (§5.2–5.4) transfer to large-scale real-world models, given the gap to the natural data?**
>
> - The controlled setup is intentionally a proof-of-concept designed to isolate the binding from confounding factors of natural data.
> - It nonetheless operates over a large scale ($1.5 \times 10^{10}$ scenes), where transformer models generalize to unseen objects.
> - The goal is to test whether a model can, in principle, learn a generalizable binding mechanism.
> - Our motivation stems from the fact that CLIP fails to generalize compositionally.
>
> **5. How do [1,2] relate to this work?**
>
> In [1,2] the authors study primarily whether object-centric inductive biases improve compositional generalization, evaluated via trained downstream models (multi-layer transformers) that use nonlinear score matching between vision and language. In contrast, we study the embedding models themselves, specifically the representations they learn during contrastive training (using cosine similarity score). [1,2] evaluate what can be done on top of frozen representations; we analyze how the representations encode binding and why they fail cross-modally. We will cite these as related work.
>
> **6. Do the conclusions extend to scenes with more objects and higher interference?**
>
>
> Our focus on 2-object scenes is intentional: CLIP already fails at binding with just 2 objects, so we focus on diagnosing this case. That said, we provide additional results for 3 objects and occlusions.
>
> We generate CLEVR scenes with 3 objects and corresponding text descriptions (**Fig. 3 in the rebuttal PDF**), and evaluate $R^2$, probing, and retrieval (Table 1 in the rebuttal PDF). The additive decomposition from §4 in the main paper extends to 3 objects: probing accuracy is 0.91–0.93 on CLEVR even with occlusions, and objects $R^2$ > concepts $R^2$.
> - We also test decomposition on natural images (**Fig. 4, Tab. 2 in the rebuttal PDF**; see also fu4L, Q1),
> - and train vision transformers from scratch with occlusions (**Figs. 1–2 in the rebuttal PDF**; see also fu4L, Q2).
>
> In both cases, our conclusions (presence of object representations, and emergence of low-complexity and multiplicative binding functions) hold.
>
>
> **7. Does the multiplicative structure also explain embeddings of models like CLIP?**
>
> CLIP fails at cross-modal binding, so we do not expect its binding function to have simple multiplicative structure. We confirm this: fitting the same multiplicative probe on CLIP and DINOv2 (**Fig. 5 in the rebuttal PDF**), concept recognition is relatively high but object recognition stays near zero.
>
> Together with the complexity and generalization results in the paper, this closes the loop: models that generalize have low-complexity multiplicative binding; CLIP does not, which explains its cross-modal failures.

---

> > ### Author Rebuttal · Reviewer_7ZcV · 2026-04-02
> >
> > Thank you for the thoughtful and detailed rebuttal, and for the additional experiments and clarifications. The rebuttal clarified the paper's intended takeaway and strengthened several parts of the submission. Based on the current rebuttal, my assessment of the paper has improved from 2 to 3. I will wait for the authors’ response to the remaining concerns before updating the score accordingly.
> >
> > **Takeaway and practical implications**:
> > Thank you for clarifying the intended main message. My remaining concern is that, while the central takeaway is now clearer, the practical implications still feel somewhat abstract to me. In particular, the paper does not yet fully show how "controlling the complexity of the binding mechanism" should translate into the design or training of real multimodal embedding models beyond the controlled setting studied here. Also related to this, I am still not sure why this representational diagnosis should matter much in practice if the relevant information already seems to be present in the embeddings and can often be extracted by relatively small and modest downstream models (e.g. [1, 2] and several works on VLMs). In that case, the issue may be less that the representation lacks the needed information, and more that the information is not organized in the specific simple form preferred by the paper. So to summarize, I am still not sure why we should care about this limitation if (1) the model already seems to contain the necessary information for the task, even if it is not directly accessible in the simplest way, (2) a relatively simple downstream model can extract this information, and (3) in many practical settings there is usually a downstream model anyway, which would in any case perform this extraction.
> >
> > **Binding-function approximation as a diagnostic for CLIP**:
> > I understand the motivation better now, but I am still not fully convinced that this diagnosis is sufficient to support the stronger claims about CLIP’s cross-modal binding failure. To me, the experiment shows primarily that the relevant structure is not easily captured by the chosen approximator family from symbolic concept specifications. I am not yet fully convinced that failure in this setup should be interpreted as directly explaining why CLIP fails cross-modally, rather than as one informative proxy for that question.
> >
> > **Meaning of “high-complexity”**:
> > I appreciate the clarification that the notion of complexity is relative to the chosen approximator family. This makes the claim more precise and reasonable. My remaining concern is mostly about the emphasis: I think the paper should be especially careful not to present this as evidence of the intrinsic complexity of the binding function, but rather as complexity relative to the specific approximation framework used in the paper.
> >
> > **Choice of approximator family**:
> > My concern in the original review was not only about the size of the MLP, but also about the choice of approximator family itself. Since the input is a discrete concept tuple, it is not obvious to me that a shallow MLP is the most suitable family for approximating the relevant mapping. It seems possible that other model classes with different inductive biases for structured or tabular inputs could perform better. If so, the current results would say more about the chosen approximator family than about the underlying complexity of CLIP’s binding map. I would therefore appreciate a bit more discussion of why this approximator family is the right one for the paper’s main claims.
> >
> > Overall, I found the rebuttal very helpful and it improved my assessment of the paper. At the same time, I still have the above concerns about the strength and phrasing of the conclusions, and I would appreciate the authors' further elaboration.

---

> > > ### Author Response · Authors · 2026-04-04
> > >
> > > Thank you for the detailed feedback.
> > >
> > > **How should controlling binding complexity translate into the design or training of real multimodal models?**
> > >
> > > Our goal is to understand how embedding models represent binding and why they fail cross-modally. That said, our results suggest concrete directions, such as regularization during contrastive training that encourages low-complexity binding (e.g. multiplicative functions), or using binding complexity as a signal for model selection (Fig. 9 via multiplicative approximations).
> > >
> > > We believe the main contribution is the mental model: binding can arise in CLS embeddings, can generalize across large scene spaces via low-complexity mappings, is accessible uni-modally in pretrained models through object structure, and show a clear gap in complexity between generalizing and non-generalizing regimes. We hope this is useful for diagnosing failures and guiding model design.
> > >
> > > **Why does this representational diagnosis matter in practice if the information is already present in the embeddings and can be extracted by small downstream models?**
> > >
> > > - Our goal is to understand why binding fails in zero-shot settings, where cosine similarity on the CLS embedding is the primary mechanism
> > > - While downstream models can recover binding, this often requires going beyond the CLS embedding. For instance, [1,2] operate on patch-level tokens rather than the global embedding, so binding is computed from spatial features. This is appropriate for downstream tasks such as VQA, but differs from binding in the CLS embedding, which determines zero-shot performance and is the level at which these models are trained.
> > > -  Even for downstream models, improving binding at the CLS level could help, e.g., by reducing the need for large data coverage as noted in [1].
> > >
> > > **Should failure in the binding approximation setup be interpreted as explaining CLIP’s cross-modal failure, or only as an informative proxy?**
> > >
> > > We agree - we do not intend to claim that any single experiment directly explains CLIP's cross-modal failure. Rather:
> > >
> > > - We observe is a consistent pattern: CLIP fails at cross-modal binding, and its binding mechanism is complex, in the sense that its MLP approximator does not generalize to unseen objects.
> > >     - Importantly, the same approximator generalizes well for concepts. CLIP has generalizable structure, the failure is specific to binding.
> > > - In contrast, our controlled transformers achieve cross-modal binding that generalizes to unseen objects, and their binding mechanism can be approximated by simple MLPs.
> > >
> > > We will adjust the language in the paper to reflect that that binding complexity provides a plausible explanation supported by multiple observations, rather than a direct explanation.
> > >
> > > **Clarify that “high-complexity” is relative to the chosen approximator, and not an intrinsic complexity of the binding function.**
> > >
> > > We will clarify in the paper that our notion of complexity is relative to the chosen approximator family and does not represent intrinsic complexity in a formal sense. By "high-complexity" we mean that the binding mechanism doesn't generalize to unseen concept combinations under this approximation framework.
> > >
> > > **Why is an MLP approximator family appropriate for the paper's claims?**
> > >
> > > Our choice of MLP is motivated by the broader view that MLPs trained with SGD tend to favor simple, compressible solutions [3,4], and compositional mappings are precisely the kind of simple solutions this bias should favor [5]. Our reasoning is that if a simple compositional binding function exists, this approximator family should find it.
> > >
> > > To verify that our findings are not specific to the MLP family, we ran additional experiments with XGBoost and Random Forest:
> > >
> > > - MLP setup is identical to §5.1. For XGBoost we use: `n_estimators` in {100, 500, 1000}, `max_depth` {6, 8, 16, 32}, `lr` in {0.1, 0.01}. For Random Forest: `n_estimators` in {100, 500, 1000}, `max_depth` in {None, 8, 16, 32}. We report best accuracy.
> > > - The result is similar to the main text: concept recognition is high while object recognition is near zero. That is, in all cases, including MLPs, the approximator succeeds at concept-level prediction but fails at object-level binding, suggesting the bottleneck is the binding structure rather than the choice of approximator.
> > > - MLPs appear to perform slightly better than XGBoost and Random forests, though this isn't an exhaustive search.
> > >
> > > | Approximator | CLIP B/32 (img) | DINOv2 B/16 (img) | CLIP B/32 (txt) |
> > > |-|-|-|-|
> > > | | Obj. / Conc. | Obj. / Conc. | Obj. / Conc. |
> > > | MLP | 0.17 / 0.97 | 0.10 / 1.0 | 0.21 / 1.0 |
> > > | XGBoost | 0.00 / 0.94 | 0.03 / 1.0 | 0.18 / 1.0 |
> > > | RF | 0.00 / 0.75 | 0.00 / 1.0 | 0.04 / 0.90 |
> > >
> > > [3] Blier, L. and Ollivier, Y. The Description Length of Deep Learning Models 2018
> > >
> > > [4] Wilson, A. G. Deep Learning is Not So Mysterious or Different 2025
> > >
> > > [5] Ren, Y. and Sutherland, D. J. Understanding Simplicity Bias towards Compositional Mappings via Learning Dynamics 2024

---

### Official Review · Reviewer_fu4L · 2026-03-13

**Soundness:** 3
**Presentation:** 4
**Significance:** 3
**Originality:** 3
**Overall Recommendation:** 5
**Confidence:** 3

**Summary:**

This paper investigates the concept binding problem in vision-language embedding models (e.g., CLIP), which refers to the ability to correctly associate features (like color) with specific objects in a multi-object scene. The authors demonstrate that while CLIP fails at cross-modal binding, it internally encodes object-level information in a clear hierarchical additive structure. Through a series of experiments, they argue that CLIP's binding failure stems from a high-complexity, non-generalizing binding function that maps concepts to objects. Finally, they show that in controlled transformer models, systematic binding generalization can emerge with sufficient data coverage, specifically characterized by multiplicative interactions between concepts.

**Compliance With Llm Reviewing Policy:**

Affirmed.

**Final Justification:**

**Soundness.** The paper presents a series of hypotheses and supports them with thorough, convincing experiments. While many experiments are conducted in simplified settings, they serve as a basis for controlled study.

**Originality.** The paper constructs a novel conceptual framework for analyzing the concept binding problem in vision-language embedding models, and it offers several original insights, particularly in framing binding as a function with varying complexity and in explaining CLIP's binding failure.

**Significance.** The concept binding problem is important yet still lacks fundamental explanations and solutions. This paper provides useful implications for improving multimodal models, although their generalizability to fully realistic settings remains to be verified in the future work.

**Clarity.** The paper is clear and well-structured, with effective visualizations.

Regarding my concerns:

W1: The rebuttal's additional experiments on more realistic data partially address the generalization issue.

W2: The added vision-based experiments alleviate concerns about reliance on synthetic inputs.

W3: The limited exploration of CLIP variants remains, but I agree with the authors that it is not central to the main contribution.

Overall, the rebuttal addressed my main concerns and strengthened my evaluation.

Final recommendation: I support acceptance. The paper provides a valuable contribution to understanding compositionality and binding in vision-language models like CLIP.

**Key Questions For Authors:**

See Weaknesses.

**Limitations:**

yes

**Strengths And Weaknesses:**

Strengths:
1. The analysis of CLIP's failure is thorough and comprehensive. The hypothesis of clear hierarchical additive structure (two-level decomposition) originates from the observation, and is validated through experiments. The idea of evaluating the complexity of the binding function is also valuable.
2. The paper develops a embedding model with low-complexity binding function and points out a key factor of the success in compositional generalization: multiplicative composition. The insight could be useful for developing better embedding models.
3. The paper is well-structured and easy to follow, with visualizations and plots clear to readers.

Weaknesses:
1. While the experiments are conducted on controlled, synthetic settings, it remains to be seen whether the takeaways hold for natural scenes, and whether the improvement can be generalized.
2. Much of the "generalization emergence" analysis (Section 5) relies on synthetic multi-hot token sequences rather than raw pixels.
3. (Minor) More CLIP variants can be examined to strengthen the claims. Is there any relationship between binding function complexity and model size?

---

> ### Author Rebuttal · Authors · 2026-03-30
>
> Thank you for the thoughtful feedback. We address the identified weaknesses below. See the [rebuttal pdf](https://anonymous.4open.science/api/repo/anon12312321321313-B64A/file/rebuttal.pdf?v=f365b) for the results.
>
> **1. "While the experiments are conducted on controlled, synthetic settings, it remains to be seen whether the takeaways hold for natural scenes, and whether the improvement can be generalized"**
>
> We agree that evaluating on natural scenes is an important direction.
>
> - Our analysis relies on controlled, balanced datasets to ensure sufficient and uniform learning signal. Such balance is typically not available in natural datasets, which makes estimating object and concept embeddings less reliable.
> - Even in the controlled setting, simple MLP approximations of the binding function do not work well. Without balanced coverage of concept combinations, learning or approximating the binding mechanism in natural scenes would be even more difficult.
>
> Nevertheless, we include results on natural images using a small, controlled dataset.
>
> We evaluate object decomposition in a more natural setting:
> - We generate images with Gemini Nano Banana 2 (gemini-3.1-flash-image-preview) (**Fig. 4 in the rebuttal PDF**), spanning 5 object types and 5 patterns (625 samples).
> - The objects exhibit variation in size, shape, color, and pattern realization, making the task substantially more challenging.
>
> **Evaluation**
> - We evaluate decomposition as in Table 1 and object editing as in Table 2 of the main text.
> - All metrics are retrieval-based:
>     - All (625): retrieve the correct scene among all generated samples.
>     - 4-way: choose among the correct option, a permuted concept combination, and two random options.
>     - Binary: choose between the original and edited scene embeddings.
>
>
> **Results**
>
> | Task            | Setting   | Retrieval | Random chance |
> |-----------------|-----------|-----------|---------------|
> | Decomposition   | All (625) | 54%       | 1/625 $\approx$ 0.16%         |
> | Decomposition   | 4-way     | 81%       | 25%           |
> | Object editing  | All (625) | 68%       | 1/625 $\approx$ 0.16%         |
> | Object editing  | Binary    | 95%       | 50%           |
>
> - Results are well above chance despite the increased difficulty.
> - This suggests that the additive structure extends, to some extent, to more natural and diverse scenes.
> - Note that this evaluation uses a relatively small dataset; with more samples, the estimated object embeddings would likely be more accurate, which could further improve results.
>
>
> **2. "Much of the "generalization emergence" analysis (Section 5) relies on synthetic multi-hot token sequences rather than raw pixels"**
>
> This was done on purpose: one of our goals was to identify what kind of mechanisms models can use to support binding in combinatorially large scene spaces (e.g. with 3 concepts, 50 values per concept, and 2 objects there are around $50^{3\cdot2} \approx 15 \times 10^9$ combinations).
>
> Nonetheless, to ensure our claims hold more broadly, we performed three experiments in vision domains that allowed us to replicate the combinatorial nature of the dataset with up to 2 concepts, including noise and overlap settings (**see Fig. 1 in the rebuttal PDF**).
>
> **Setup**
> - We adapted the transformer encoder architecture by replacing the multi-hot input layer with a convolutional front-end and increasing the number of layers from 6 to 8 (since this lead to models generalizing better).
> - The rest of the pipeline (training, evaluation, binding analysis) remains identical to the main paper.
> - We consider three levels of visual complexity: (i) noise-free, non-overlapping objects, (ii) speckled noise with non-overlapping objects, and (iii) noisy and overlapping objects.
> - Each object is defined by two concepts (square color and border color), with $k=2, n=50$, leading to up to $6.5 \times 10^6$ possible object combinations.
>
> **Results**
>
> The findings from the main paper replicate in the vision setting (**Fig. 2 in the rebuttal PDF**):
> - The binding map of models that generalize is well approximated by shallow MLPs, with concept-level reconstruction performing well.
> - Models that generalize exhibit multiplicative structure, consistent with our findings on multi-hot inputs.
>
>
> **3. "(Minor) More CLIP variants can be examined to strengthen the claims. Is there any relationship between binding function complexity and model size?"**
>
> Thank you for the suggestion. Examining more CLIP variants and how binding complexity relates to model size is an interesting direction. It is not the main focus of this work, which centers on characterizing the binding mechanism itself, but we agree it would be valuable to investigate further.

---

> > ### Author Rebuttal · Reviewer_fu4L · 2026-04-03
> >
> > Thank you for your detailed response. My concerns have been addressed by the additional results, and I would like to maintain my original score, which is already good.

---

### Official Review · Reviewer_EmAT · 2026-03-13

**Soundness:** 4
**Presentation:** 4
**Significance:** 3
**Originality:** 3
**Overall Recommendation:** 5
**Confidence:** 3

**Summary:**

This paper attempts to understand why vision–language embedding models don't represent a binding between the objects in a scene and their corresponding concepts. To do so, this paper analyzes CLIP/DINO image and text representations of scenes with N objects (each corresponding to a set of concepts, e.g. shape, color).

In particular, they find that (1) scene embeddings do decompose into the sum of object representations and (2) object representations don't decompose into the sum of concept representations. They find evidence for (1) but not (2), which supports the idea that the object–concept correspondence is not simply (i.e. additively) represented in the embedding space. Next, they approximate this binding function (from concept to scene representations) using an MLP  and argue that this MLP requires higher capacity (pointing to models' tendency for memorization, which explaining failures in generalization).

Finally, they train Transformer models in a synthetic setting and find that more general (and more systematic) binding functions are learned with sufficient data coverage.

**Compliance With Llm Reviewing Policy:**

Affirmed.

**Final Justification:**

I found that this was a well-written and well-designed paper that investigates an important problem (the binding problem in vision–language models). The authors resolved any questions I had in their rebuttal. I would vote to Accept this paper.

**Key Questions For Authors:**

I just have some advice and clarification questions for the authors. Apologies if I missed or misunderstood something.

- I think it might be confusing to write "CLIP's binding" or "CLIP's binding function", since what you are referring to isn't an explicit "function" in the model (rather something implicitly learned). E.g. for L18 (col 2), maybe "how objects and concepts are geometrically related in CLIP's embedding space"?

- I think it would be helpful to clarify what "high-complexity" means when it is first mentioned. Also, why is it that "if both binding functions have low-complexity, they are more likely to produce aligned embeddings" (L139, col 2)?

- I think Figure 2 is pretty hard to understand. Maybe it's better to lean more on the presented formula.

- Is the hypothesis that the scene embedding decomposes exactly into the sum of object representations? I think that would be an oversimplification, since we might expect other information to be surfaced in the scene embedding (just as example: the positions of the objects or the number of objects, but also more abstract scene-level concepts). Anyway you find a residual in practice, in Sec. 4.1, right?

- Can you emphasize what $u_o$ is earlier and how it is precisely computed? Moreover, I couldn't find a definition for $u_c$.

- What are you fitting linear probes on in Sec. 4.1 (L250-252)? Predicting the residual from $f(x_s)$?

- Can you offer a formula in Sec. 4.2 to illustrate the experiment, like in the previous section?

**Limitations:**

I couldn't find any discussed limitations. Could the authors consider this further? Authors also state that there could be societal impact from general efforts of advancing machine learning, but none that are specific to this work.

**Strengths And Weaknesses:**

**Presentation:** This is a very well written paper, which properly engages with the prior work, and many clear definitions and experimental designs.

**Soundness:** I think the experimental design is clear and appears sound to me. (I have minor clarification questions in the subsequent section.) Each research question is tested with multiple experiments, two model families are tested, multiple metrics are reported. This paper includes both a representational analysis of pre-trained models and controlled analyses of (500) models trained from scratch. I believe this is sufficient experimental coverage for a research paper.

**Significance / originality:** This paper tries to explain a common observation in vision–language models using simple and intuitive framework of learned representations. They present a clear finding that scene representations additively decompose to their object representations. There is more than sufficient interest in binding and compositional generalization in vision–language models, so I think this is an important contribution to the community. An important (perhaps unstated) implication of this work is that models like CLIP require greater data coverage than their (already vast) training data offers. This presents a call for architectures and learning methods that systematically generalize with more data efficiency.

Critics would pick on the toy nature of datasets and concepts tested in this work. It's true that this work doesn't offer a complete explanation of the capabilities of vision–language models. However, this work sufficiently measures a fundamental capability in controlled settings --- I think it is hard to ask for more than that.

---

> ### Author Rebuttal · Authors · 2026-03-30
>
> Thank you for the thoughtful feedback.
>
> **1. "it might be confusing to write "CLIP's binding" or "CLIP's binding function" [...] E.g. for L18 (col 2), maybe "how objects and concepts are geometrically related in CLIP's embedding space"?"**
>
> We agree that "CLIP's binding" at L18, col. 2 may be confusing, especially early in the introduction; we will clarify it in the revision.
>
> **2. "it would be helpful to clarify what "high-complexity" means when it is first mentioned."**
>
> We agree that this should be clarified earlier. Our notion is motivated by complexity-based views of compositionality: if the concept-to-scene map follows a simple reusable rule, then a simple approximator should be able to learn it and generalize. Since Kolmogorov complexity is intractable, we operationalize this using a restricted approximator family of 1-hidden-layer MLPs of varying width. We will clarify this in the first mention (L30, col. 2).
>
> **3. "why is it that "if both binding functions have low-complexity, they are more likely to produce aligned embeddings" (L139, col 2)?"**
>
> This is motivated by Occam's razor: among functions consistent with observed data, those with simpler, more reusable structure are more likely to generalize beyond the training domain [1].
> - If the vision and language modalities are aligned on a domain $\mathcal{D}$ through functions $f$ and $g$, and both are low-complexity, then there are fewer possible mappings consistent with $\mathcal{D}$, making it more likely that both modalities converge on the (low complexity and reusable) rule. This alignment is therefore more likely to extend outside $\mathcal{D}$ when the shared rule they implement is simple and compositional.
> - Intuitively, alignment requires $f$ and $g$ to produce similar representations even for unseen objects. If the binding function is complex, this is difficult: there are many combination-specific ways to fit the observed domain without agreeing off-domain. If instead a simple reusable rule is used (e.g., a multiplicative mechanism), then alignment is more likely because both modalities can rely on the same shared mechanism.
>
> [1] Elmoznino, Eric, et al. "A complexity-based theory of compositionality." arXiv:2410.14817 (2024).
>
> **4. "Figure 2 is pretty hard to understand."**
>
> We agree that Fig. 2 contains many details, which may make it hard to follow. The goal of this visualization is to provide intuition for the additive decomposition: scenes with two objects lie between the embeddings of the corresponding single-object scenes, which is consistent with the additive model in Eq. 4.
>
> We will improve the clarity of both the figure and its explanation in the revision.
>
> **5. "Is the hypothesis that the scene embedding decomposes exactly into the sum of object representations?**
>
> We agree that exact decomposition into a sum of object representations would be an oversimplification. In practice, the reconstruction does not explain all variance, as reflected by the $R^2$ values in Table 1 (Sec. 4.1), indicating a residual that may encode additional scene details. We will make this clearer in the revision.
>
> **6. Clarify the definitions and computation of $u_o$ and $u_c$.**
>
> Thank you. We will add clear definitions at the beginning of Sec. 4.
>
> - *Object embeddings* $u_o$ are estimated as averages of scene embeddings over scenes that contain object $o$. Let $\mathcal D_o := \lbrace s \in \mathcal{S} : o \in O(s) \rbrace$ be the set of scenes containing object $o$. Then
> $$
> \hat u_o = \mathbb{E}_{s \sim \mathcal{D}_o} [ f(x_s) ]
> $$
> That is, the embedding for a given object (e.g., "red square") is obtained by averaging the embeddings of all scenes containing that object, regardless of what other objects appear or in what position.
>     - We also consider position-specific and single-object variants, described in Appendix D and E.
>
> - *Concept embeddings* $u_c$ are defined analogously by averaging over scenes containing concept value $c$. Let $\mathcal D_c := \lbrace s \in \mathcal S : c \in V_i(s) \rbrace$. Then
> $$
> \hat u_c = \mathbb{E}_{ s \sim \mathcal D_c } [ f(x_s) ]
> $$
> For example, the embedding for "red" is obtained by averaging the embeddings of all scenes containing a red object.
>
> **7. "What are you fitting linear probes on in Sec. 4.1 (L250-252)?"**
>
> The linear probes in Sec. 4.1 are trained on the residual, i.e., the part of the embedding remaining after subtracting concept or object components from $f(x_s)$. They predict the concept and object content of the input embedding (as in Defns. 3.3, 3.4).
>
> **8. "Can you offer a formula in Sec. 4.2?"**
>
> We will clarify that the formula used in Sec. 4.2 is the same as in Eq. 4, with scene embeddings decomposed into concept-level sums per object:
> $$
> f(x_s) = \sum_{i=1}^{C} u_{c_1,i} + \sum_{i=1}^{C} u_{c_2,i}
> $$
> where $u_{c_k,i}$ denotes the embedding of the $i$-th concept for object $k$. For example, for "red cube and blue sphere", we sum the embeddings of "red", "cube", "blue", and "sphere".

---

> > ### Author Rebuttal · Reviewer_EmAT · 2026-04-03
> >
> > Thanks to the authors for answering all my questions. I am satisfied with their responses and I also checked the other reviews, I am not deterred by the "Reject" review. I will maintain my original score, which was already good.

---

### Decision · Program_Chairs · 2026-04-30

**Decision:**

Accept (spotlight)

**Comment:**

The paper proposes a novel approach to decomposing embeddings into interpretable concepts, supported by rigorous analysis that yields new insights into the compositional structure of multimodal embeddings via binding. Following the rebuttal, all reviewers converged toward a consensus in favor of acceptance.

That said, some minor concerns remain. For instance, Reviewer 7ZcV noted that the practical implications are somewhat abstract and that there were also minor comments regarding the phrasing of the conclusions.

Overall, these weaknesses are limited in scope and do not detract from the paper’s strong technical contributions and insights. I therefore recommend acceptance. The authors are strongly encouraged to carefully incorporate the reviewers’ feedback during the camera-ready preparation.